# YAP/TAZ initiate and maintain Schwann cell myelination

Matthew Grove[1,2], Hyukmin Kim[1,2], Maryline Santerre[3], Alexander J Krupka[4], Seung Baek Han[1,2], Jinbin Zhai[1,2], Jennifer Y Cho[1], Raehee Park[1,2], Michele Harris[2], Seonhee Kim[1,2], Bassel E Sawaya[3], Shin H Kang[1,2], Mary F Barbe[2], Seo-Hee Cho[1,2], Michel A Lemay[4], Young-Jin Son[1,2]*

[1]Shriners Hospitals Pediatric Research Center, Center for Neural Repair, Lewis Katz School of Medicine, Temple University, Philadelphia, United States; [2]Department of Anatomy and Cell Biology, Lewis Katz School of Medicine, Temple University, Philadelphia, United States; [3]FELS Institute for Cancer Research and Molecular Biology, Lewis Katz School of Medicine, Temple University, Philadelphia, United States; [4]Department of Bioengineering, Temple University, Philadelphia, United States

**Abstract** Nuclear exclusion of the transcriptional regulators and potent oncoproteins, YAP/TAZ, is considered necessary for adult tissue homeostasis. Here we show that nuclear YAP/TAZ are essential regulators of peripheral nerve development and myelin maintenance. To proliferate, developing Schwann cells (SCs) require YAP/TAZ to enter S-phase and, without them, fail to generate sufficient SCs for timely axon sorting. To differentiate, SCs require YAP/TAZ to upregulate Krox20 and, without them, completely fail to myelinate, resulting in severe peripheral neuropathy. Remarkably, in adulthood, nuclear YAP/TAZ are selectively expressed by myelinating SCs, and conditional ablation results in severe peripheral demyelination and mouse death. YAP/TAZ regulate both developmental and adult myelination by driving TEAD1 to activate Krox20. Therefore, YAP/TAZ are crucial for SCs to myelinate developing nerve and to maintain myelinated nerve in adulthood. Our study also provides a new insight into the role of nuclear YAP/TAZ in homeostatic maintenance of an adult tissue.

*For correspondence: yson@temple.edu

**Competing interests:** The authors declare that no competing interests exist.

## Introduction

Normal motor and sensory functions depend on myelination of peripheral axons by myelin-forming Schwann cells (ie., myelinating SCs; mSCs) that enables fast neural transmission. Failure to generate or maintain mSCs leads to life-threatening peripheral neuropathy and myelinopathy. Research over the last few decades has made great progress in identifying extra- and intracellular signals that drive SC differentiation and myelination during development, but mechanisms that maintain mSCs and myelination in adults remain unclear (recent reviews in *Jessen et al., 2015*; *Salzer, 2015*; *Monk et al., 2015*; *Taveggia, 2016*). Krox20 (also called Egr2) has been identified as the master transcription factor that drives myelin gene expression for both myelin formation and maintenance (*Topilko et al., 1994*; *Decker et al., 2006*). However, it remains unclear how extra- and intracellular signals regulate transcription of Krox20. A major gap in our knowledge concerns the nucleocytoplasmic mediator(s) that converge and deliver extra- and intracellular signals to the SC nucleus to drive Krox20. Although several transcription factors upstream of Krox20 have been identified, they either reside only in the nucleus or play uncertain roles in Krox20 regulation (*Stolt and Wegner, 2016*).

YAP (Yes-associated protein) and its paralogue, TAZ (Transcriptional coactivator with PDZ-binding motif), are Hippo pathway transcriptional co-activators and potent oncoproteins that shuttle

between cytoplasm and nucleus (*Hansen et al., 2015*). When activated, YAP and TAZ (hereafter YAP/TAZ) translocate to the nucleus, where they potently induce cell proliferation. Because YAP/TAZ lack DNA binding domains, they act indirectly by binding partner transcription factors, particularly members of the TEA domain (TEAD) family (*Zhao et al., 2008*). Following phosphorylation by activated Hippo pathway, YAP/TAZ are inactivated through cytoplasmic retention and ubiquitin-mediated degradation. This nucleocytoplasmic shuttling of YAP/TAZ is critical for developmentally regulating cell proliferation that ensures normal tissue growth and organ size (*Piccolo et al., 2014*). YAP/TAZ shift to the cytoplasm concomitant with differentiation and the nuclear exclusion of YAP/TAZ is a requisite for adult tissue homeostasis (*Varelas, 2014*). Indeed, nuclear translocation of YAP/TAZ in differentiated adult cells is essential for cancer initiation or growth of most solid tumors (*Harvey et al., 2013*; *Moroishi et al., 2015*). Accordingly, YAP/TAZ are emerging as attractive targets for cancer therapies (*Zanconato et al., 2016*). YAP/TAZ also interact with other signaling pathways, including Wnt, Notch, GPCR, and mechanical signals (*Yu et al., 2012*; *Azzolin et al., 2012*; *Rayon et al., 2014*; *Heallen et al., 2011*). These findings suggest that nuclear YAP/TAZ have additional context-dependent functions besides their widely recognized role in promoting cellular proliferation and developmental differentiation.

Studies from three laboratories have recently implicated YAP/TAZ in myelination of developing peripheral nerve. Poitelon et al. showed that conditional knockout mice lacking YAP/TAZ in SCs are peripherally unmyelinated, as these SCs completely fail to sort out large-caliber axons. This is a prerequisite step in developmental myelination known as radial or axon sorting. The authors observed no nuclear YAP/TAZ in cultured myelinating SCs, and modest, if any, regulation of Krox20 by YAP/TAZ (*Poitelon et al., 2016*). Two subsequent studies raised the possibility that YAP/TAZ may regulate myelination more directly: these reports showed upregulation of myelin genes, including Krox20, by TEAD1 (*Lopez-Anido et al., 2016*) or YAP (*Fernando et al., 2016*). In addition, Fernando et al. reported that viral inactivation of YAP in developing sciatic nerve reduces SC and myelin length, although myelin thickness remains unaltered (*Fernando et al., 2016*). How YAP/TAZ regulate developmental myelination is therefore unclear, and whether they also play a role in maintenance of myelination in adulthood is completely unknown.

In this report we first show that SCs lacking YAP/TAZ are in fact capable of sorting axons, but the process is extensively delayed and incomplete, largely due to defective SC proliferation. Sorted axons never became myelinated, however, apparently due to the failure of these SCs to upregulate Krox20 and differentiate into mSCs. We also show that adult mSCs, but not non-myelinating SCs, express nuclear YAP/TAZ, and that induced ablation of YAP/TAZ in adult SCs results in Krox20 downregulation, overt demyelination and, notably, mouse death. Lastly, we report that a YAP/TAZ-TEAD1 complex regulates both developmental and adult myelination, presumably through transcriptional regulation of Krox20. Our results clarify a developmental function of SC YAP/TAZ, and newly establish YAP/TAZ as essential regulators of both developmental and adult myelination in the peripheral nervous system. Our study also calls for caution in targeting YAP/TAZ for cancer therapy.

## Results

### YAP and TAZ are expressed in proliferating, differentiating and mature myelinating Schwann cells

We began our study by testing a number of commercial antibodies for their specificity to YAP and its paralogue, TAZ. Using conditional knockout (cKO) mouse lines that selectively lack YAP, TAZ, or both YAP/TAZ in SCs, we identified an antibody that specifically recognized both YAP and TAZ (*Figure 1—figure supplement 1A and B*), and antibodies that bound to only YAP (*Figure 1—figure supplement 1A and C*). We also identified an antibody specific for phosphorylated YAP (p-YAP) preferentially present in cytoplasm (*Figure 1—figure supplement 1A and D*, and see also *Figure 1B*). During the antibody screening, we found that adult SCs expressed both YAP and TAZ, because YAP/TAZ immunoreactivity was eliminated in *Yap/Taz* double cKO (cDKO), but not in *Yap* or *Taz* single cKO (*Figure 1—figure supplement 1B*, see also *Figure 2—figure supplement 1A and B*).

Using these antibodies and an antibody for Sox10, a specific marker of the SC nucleus, we confirmed that YAP/TAZ were strongly localized in the nuclei of immature SCs at E16.5 (*Figure 1A and E*), when SCs are actively proliferating and sorting out large axons (*Jessen and Mirsky, 2005*).

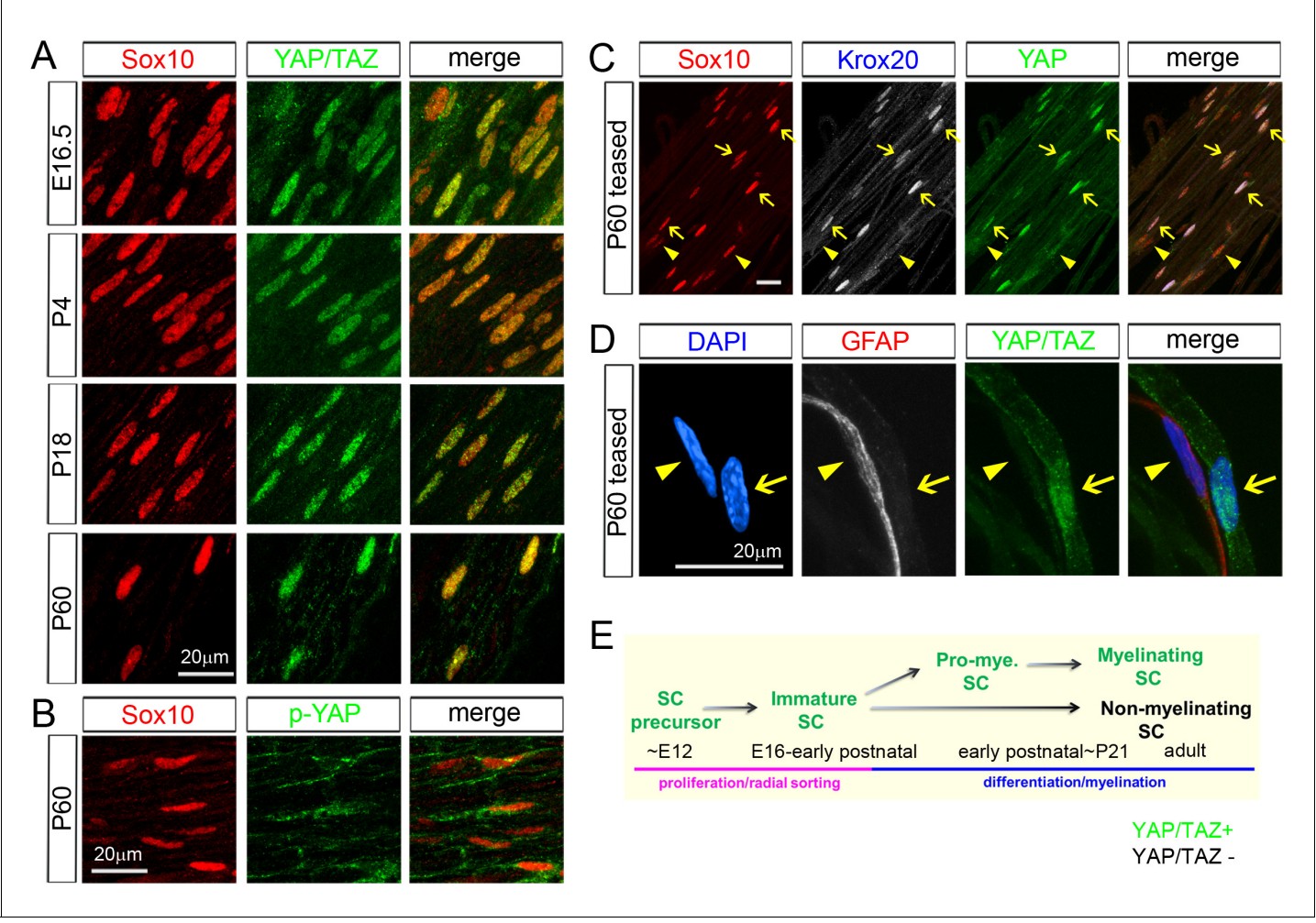

**Figure 1.** YAP/TAZ are largely nuclear in proliferating, differentiating and mature myelinating SCs. YAP/TAZ localization was investigated in sciatic nerve cryosections (A, B) or teased fibers (C, D). SC nuclei are marked by Sox10 (red) or DAPI (blue). (A) YAP/TAZ (green) are in SC nuclei throughout development, from E16.5 to P60. (B) YAP phosphorylated on Ser 112 (p-YAP, green) is largely absent from SC nuclei at P60. (C) Co-localization of YAP (green) in the nuclei of myelinating SCs marked by Krox20 (white) at P60. Arrows point to mSC nuclei that contain YAP and Krox20; arrowheads point to non-myelinating SC nuclei, which lack Krox20 and YAP. (D) YAP/TAZ (green) are present in the nucleus of a mSC, associated with a large axon, but is absent from the nucleus of a non-myelinating SC, marked by GFAP (red), at P60. Arrow points to myelinating SC nucleus; arrowhead points to non-myelinating SC nucleus. (E) Diagram summarizing the developmental stages in which YAP/TAZ are present or absent from the SC nucleus. The following figure supplements are available for *Figure 1*.

The following figure supplement is available for figure 1:

**Figure supplement 1.** Selection of specific antibodies for immunohistochemical analysis of YAP and TAZ.

Notably, YAP/TAZ continued to be expressed strongly in SC nuclei at P4, P18 and P60. YAP/TAZ have distinct functions when they are in the cytoplasm versus in the nucleus (*Diepenbruck et al., 2014*; *Varelas, 2014*): YAP/TAZ actively regulate transcription when they are localized to the nucleus. In contrast, when YAP/TAZ are phosphorylated and in the cytoplasm, they are presumed to be transcriptionally inactive. This result suggest, therefore, that YAP/TAZ act as transcriptional regulators not only in proliferating and differentiating SCs, but also surprisingly in fully mature SCs.

Immunostaining of transcriptionally inactive, phosphorylated-YAP (p-YAP) was largely cytoplasmic, as expected, indicating nucleocytoplasmic shuttling of YAP/TAZ in mature SCs (*Figure 1B*). Notably, not all postnatal SCs expressed YAP/TAZ: they were expressed by ~60% at P4 and ~75% at P60, approximately corresponding to the percentage of myelinating SCs (mSCs; *Figure 2—figure*

supplement 1C). We therefore co-immunostained P60 teased sciatic nerve fibers for Sox10, Krox20 (to selectively label mSCs) and YAP. Strikingly, YAP was highly expressed in mSCs (*Figure 1C*; arrows), but virtually undetectable in Krox20-negative, non-myelinating SCs (*Figure 1C*; eg., arrowheads). We verified this result by co-immunostaining teased fibers for YAP/TAZ and for GFAP, which exclusively labels non-myelinating SCs (*Figure 1D*; *Jessen et al., 1990*) (*Figure 1D*). These results show that YAP/TAZ are actively regulating transcription in proliferating and differentiating SCs during development and then selectively in differentiated mSCs in the adulthood (*Figure 1E*). Selective expression of YAP/TAZ in mSCs raise the possibility that YAP/TAZ are directly involved in transcriptional regulation of SC differentiation and myelination.

## YAP/TAZ are required for myelination of sorted axons

To investigate the roles of YAP/TAZ in developing SCs, we conditionally ablated YAP, TAZ or both selectively in SCs by breeding $Yap^{fl/fl}$ and $Taz^{fl/fl}$ mice (*Xin et al., 2013*) with mice carrying *P0-Cre*. P0-Cre is specifically active in SCs from E13.5, when SCs are at the precursor stage (*Feltri et al., 1999*). We also generated *Yap* cKO heterozygous for *Taz* ($Yap^{fl/fl}$; $Taz^{fl/+}$; *P0-Cre*, hereafter *Yap*-cKO/*Taz*-cHET) and *Taz* cKO heterozygous for *Yap* ($Taz^{fl/fl}$; $Yap^{fl/+}$; *P0-Cre*, hereafter *Taz*-cKO/*Yap*-cHET). Biochemical and immunohistochemical analysis of the *Yap* cKO, *Taz* cKO and *Yap/Taz* double cKO mice (hereafter *Yap/Taz* cDKO) showed efficient Cre-mediated deletion that was selective for SCs (*Figure 2—figure supplement 1A and B*). In our rigorous quantitative analysis, ~20% SC nuclei were YAP/TAZ+ in P20 cDKO sciatic nerves and the number was not increased at P60 (*Figure 2—figure supplement 1C*, and see below); notably, many of this 20% exhibited only faint or low immunoreactivity, RT-qPCR analysis of sciatic nerve RNA also showed marked reduction of *Yap/Taz* mRNA expression in cDKO mice (*Figure 2—figure supplement 1D*), though as expected, significant *Yap/Taz* mRNAs were still detected, most likely due to non-SCs which strongly expressed YAP/TAZ, such as perineurial cells, fibroblasts and presumably vascular cells (eg., asterisks in *Figure 2—figure supplement 1B*).

Consistent with a recent report (*Poitelon et al., 2016*), postnatal *Yap/Taz* cDKO developed severe peripheral neuropathy, whereas neither *Yap* nor *Taz* cKO showed an abnormal phenotype. As early as P7, cDKO mice began to develop tremor, splayed gait and hindlimb paralysis, which progressively worsened with age (data not shown). We examined *Yap/Taz* cDKO mice throughout adulthood, in contrast to Poitelon et al., who analyzed them up to P20 (*Poitelon et al., 2016*). By P60 both hindlimbs were completely paralyzed, and the forelimbs displayed obvious weakness (*Figure 2A*, *Video 1*). We euthanized cDKO mice for humane reasons at ~P90 and focused our initial study on P60 cDKO mice. Behavioral analyses of adult cDKO mice showed severely impaired motor and sensory function in forelimbs and hindlimbs (*Figure 2—figure supplement 2*). Sciatic nerves of P60 *Yap/Taz* cDKO were translucent due to virtually complete absence of myelin (*Figure 2B*), and compound muscle action potentials (CMAPs) were absent in footpad muscles. Much stronger stimulation elicited CMAPs only infrequently, and amplitudes were markedly delayed and dispersed (*Figure 2C*). Notably, spontaneous and irregular potentials occurred in P60 cDKO footpad muscles (*Figure 2D*). These spontaneous potentials are pathological and were likely generated by the abundant, abnormal extrasynaptic contacts in cDKO muscles that resulted from extensive intramuscular sprouting (for example, *Figure 4—figure supplement 3B and C*).

Cellular analysis confirmed complete amyelination in P60 cDKO mice. In stark contrast to controls, myelin basic protein (MBP) was undetectable (*Figure 2E1*), except in the rare SCs which retained strong YAP/TAZ immunoreactivity (*Figure 2E1 and E*, arrows). Those SCs exhibiting low or faint YAP/TAZ immunoreactivity did not express MBP (for example, #2 SC in *Figure 2E2*). Semithin sections and electron microscopy provided additional evidence of the nearly complete absence of myelin (*Figure 2F and G*). Notably, however, although we frequently observed small bundles of unsorted axons, many axons were singled out by promyelinating SCs with thick basal lamina in P60 cDKO (*Figure 2F* and inset). Similarly, axons in forelimb nerves, dorsal roots and ventral roots in P60 cDKO exhibited complete amyelination, and frequently established a 1:1 relationship with promyelinating SCs (*Figure 2—figure supplement 3D*). We frequently observed small bundles of unsorted axons and occasionally large axon bundles (*Figure 2—figure supplement 3E*), but we did not observe polyaxonal myelination in which multiple axons were surrounded by a mSC (*Figure 2—figure supplement 3D*, arrows). These results demonstrate that SCs lacking YAP/TAZ arrest as promyelinating SCs (that is, fail to differentiate into mSCs), and are therefore unable to myelinate axons.

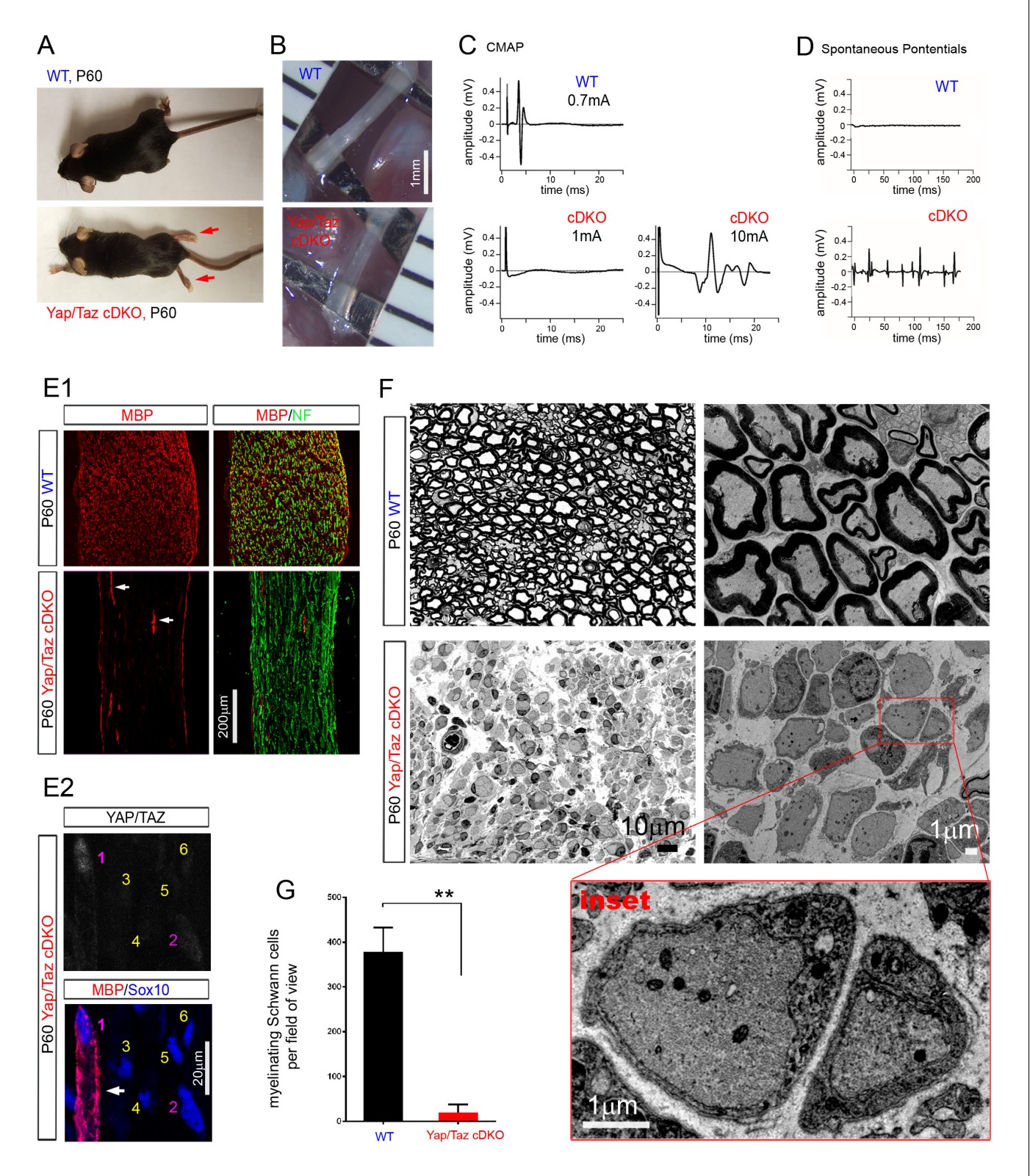

**Figure 2.** YAP/TAZ are required for myelination of sorted axons. (**A**) Living control and *Yap/Taz* DKO mice, showing splayed paralyzed hindlimbs (arrows) and muscle wasting of P60 *Yap/Taz* cDKO mice. (**B**) Control sciatic nerve is white and opaque, while cDKO sciatic nerve is translucent. (**C**) CMAPs generated by stimulation of the sciatic nerve. (**D**) Spontaneously generated abnormal potentials in cDKO footpad muscles. (**E1, E2**) Low and high magnification views of longitudinal sections of WT and P60 cDKO sciatic nerves. Arrows denote mSCs exhibiting strong MBP and YAP/TAZ

*Figure 2 continued on next page*

*Figure 2 continued*

immunoreactivity. (**E1**) cDKO nerves rarely contain mSCs (arrows). mSCs are marked with MBP (red), and axons are marked with neurofilament (green). (**E2**) An example of an mSC (ie, MBP+) in P60 cDKO, which shows strong YAP/TAZ immunoreactivity (#1 SC; arrow). Note that adjacent SCs show low (#2 SC) or no YAP/TAZ immunoreactivity (#3–6 SCs), and they do not express MBP. (**F**) Left half panels: semi-thin transverse sciatic nerve sections showing mSCs in control but not in cDKO. Right half panels: TEM of transverse nerve sections showing myelination of large axons in control (upper panel), but many large axons arrested at the 1:1 promyelinating stage in cDKO. Lower panel: Inset of the same cDKO panel, showing two large axons surrounded by cDKO SCs arrested at the promyelinating stage. (**G**) Quantification of myelinating Schwann cells per field of view in WT and cDKO semi-thin sections. n = 3 mice per genotype. **p=0.0034, unpaired Student's t-test. The following figure supplements are available for *Figure 2*.

The following figure supplements are available for figure 2:

**Figure supplement 1.** Recombination efficiency in *Yap* cKO, *Taz* cKO and *Yap/Taz* cDKO.
**Figure supplement 2.** Behavioral tests of motor and sensory function of adult *Yap/Taz* cDKO mice.
**Figure supplement 3.** Complete amyelination of forelimb peripheral nerves and nerve roots in P60 *Yap/Taz* cDKO.

In contrast to *Yap/Taz* cDKO, myelination appeared completely normal in the sciatic nerves of P60 *Yap* cKO (*Figure 2—figure supplement 3B*) and nearly so in those of P60 *Taz* cKO, except for rare small bundles of unsorted axons (*Figure 2—figure supplement 3C*). Sciatic nerves of P60 *Yap*-cKO/*Taz*-cHET appeared fully myelinated, whereas those of P60 *Taz*-cKO/*Yap*-cHET exhibited large bundles of unsorted axons, in addition to normal and polyaxonal myelination (*Figure 3B*). These observations were consistent with the behavioral phenotypes of the mice. *Yap*-cKO/*Taz*-cHET mice appeared normal, whereas *Taz*-cKO/*Yap*-cHET showed mild tremor, wide gait and progressive paralysis in adulthood. Unlike P60 cDKO mice, however, *Taz*-cKO/*Yap*-cHET mice walked on their hindlimbs, and hindlimb paralysis never became complete (data not shown). These results show that YAP and TAZ are required for SC differentiation and myelination. They also suggest that YAP/TAZ are redundant in regulating SC differentiation and myelination, and that TAZ is the more potent regulator.

## YAP/TAZ are required for timely axon sorting

Our observation of many sorted axons in P60 cDKO contradicts a recent report of a complete failure of radial sorting in *Yap/Taz* cDKO mice (*Poitelon et al., 2016*). This is an important issue because incomplete but extensive axon sorting would suggest direct regulation of SC myelination by YAP/TAZ, whereas the complete failure would suggest that regulation is indirect and consists of blocking radial sorting, a pre-requisite step in SC differentiation and myelination. We therefore analyzed developmental patterns of radial sorting and SC differentiation in *Yap/Taz* cDKO mice.

At P4, cDKO sciatic nerves contained large bundles of unsorted, mixed-caliber axons, whereas many axons were already sorted and myelinating in controls (*Figure 3A,P4*). At P18, when myelination appeared complete in control nerves, proximal nerves of cDKO contained many large bundles of unsorted axons (*Figure 3A*: P18 proximal). Notably, however, the unsorted axon bundles were appreciably smaller in distal nerves than in proximal nerves (*Figure 3A,P18* distal), and axons 1:1 with promyelinating SCs were present (*Figure 3A,P18* EM). At P60, many axons were sorted but remained unmyelinated in proximal and distal nerves of cDKO, although small bundles of unsorted axons were frequently observed (*Figure 3A,P60*; see also *Figure 2—figure supplement 3E*). We also observed small caliber axons fully surrounded by single non-myelinating SCs (*Figure 3A*; marked by 'a' and 'SC'). These results show that radial sorting is not blocked in *Yap/Taz* cDKO mice, but that it proceeds abnormally and remarkably slowly.

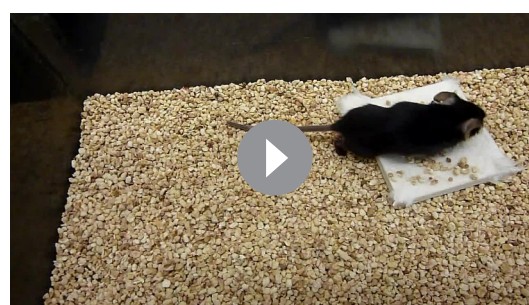

**Video 1.** A movie showing a P60 *Yap/Taz* cDKO mouse.

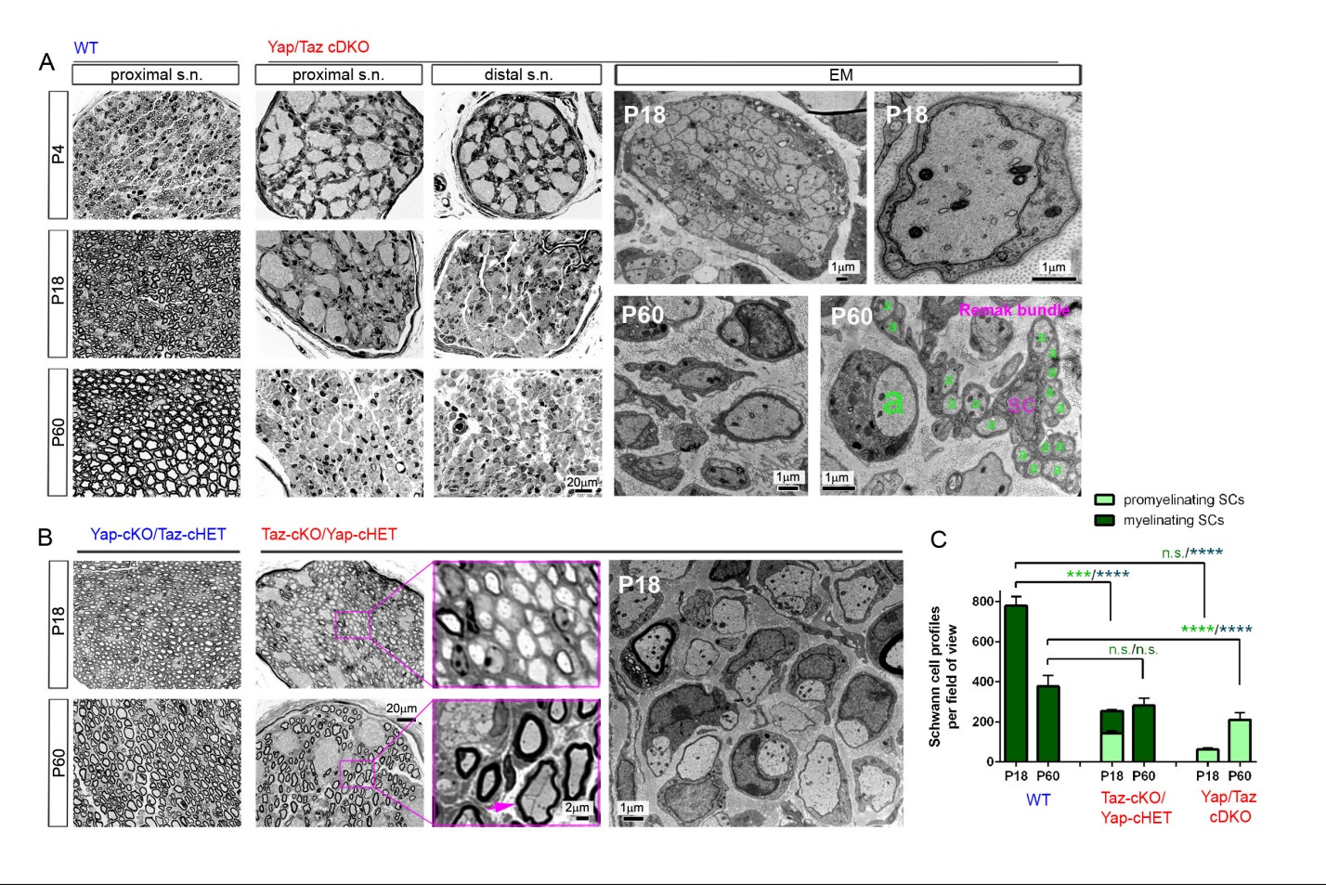

**Figure 3.** YAP/TAZ are required for timely axon sorting. (A) Left nine panels: Semi-thin transverse sections of proximal control sciatic nerve and proximal and distal cDKO nerves at P4, P18 and P60. Right four panels: TEM of transverse cDKO nerve sections at P18 and P60. At P18, some immature SCs associate with unsorted axon bundles (left panel), but other SCs are promyelinating (right panel). At P60, many large axons are sorted but arrested at the 1:1 promyelinating stage (left and right panels, marked by large 'a'), while small axons are fully surrounded by the cytoplasm of non-myelinating SCs in extended Remak bundles. s.n. = sciatic nerve; a = axon; SC = non-myelinating SC. (B) Transverse sections of *Yap*-cKO/*Taz*-cHET and *Taz*-cKO/*Yap*-cHET sciatic nerves at P18 and P60. Left-most two small panels: Semi-thin sections of *Yap*-cKO/*Taz*-cHET. Right-most four small panels: Semi-thin sections of *Taz*-cKO/*Yap*-cHET. Large panel: TEM of *Taz*-cKO/*Yap*-cHET, showing single large axons associated with SCs transiently arrested at the promyelinating stage. (C) Quantification of myelinating vs promyelinating Schwann cells in transverse semi-thin sections of WT vs *Taz* cKO/*Yap* cHET and *Yap/Taz* cDKO sciatic nerves at P18 and P60. n = 3 mice per genotype, except for WT P60 (n = 2). P value significance is shown in pale green for promyelinating Schwann cells and in dark green for myelinating Schwann cells. ***p<0.001, ****p<0.0001, n.s. = non-significant, 2-way ANOVA with Sidak's multiple comparison test.

The discrepancy between our and Poitelon et al.'s studies is likely due to the fact that Poitelon et al. examined *Yap/Taz* cDKO mice up to P20 (*Poitelon et al., 2016*), whereas we analyzed them more thoroughly up to P60. Several considerations excluded the possibility that delayed axon sorting in our cDKO mice was due to incomplete deletion of *Yap/Taz*. First, we used the same *P0 Cre*-driver line as Poitelon et al. and observed a major block in radial sorting at P18. Second, if an allele of *Yap* or *Taz* remained undeleted, these SCs would have differentiated to mSCs and myelinated axons by P18, as in control nerves. However, we found mSCs to be rare in P18 and P60 cDKO (for example, MBP+ and YAP/TAZ+ SCs in cDKO in *Figure 2E2*; *Figure 3C*). Third, not more than ~20% SCs in cDKO were immunopositive for YAP/TAZ at P60, which is too few to explain the incomplete but extensive radial sorting and virtually complete amyelination (see below). Fourth, most of these ~20% SCs were only weakly immunopositive at P4 and did not increase at P60, indicating that they failed to proliferate vigorously. Notably, whereas an allele of *Taz* was sufficient to

complete myelination in P60 *Yap*-cKO/*Taz*-cHET (*Figure 3B*; leftmost panels), an allele of *Yap* was insufficient as evidenced by defective sorting and myelination in P60 *Taz*-cKO/*Yap*-cHET (*Figure 3B*). We therefore postulate that most of the weak *Yap/Taz* immunoreactivity in these ~20% SCs represents an undeleted allele of *Yap* or non-specific antibody binding.

We frequently observed promyelinating SCs that sorted but did not myelinate axons at P18 in *Taz*-cKO/*Yap*-cHET mice (*Figure 3B and C*). These promyelinating SCs differentiated into mSCs, myelinated axons and disappeared by P60 (*Figure 3C*). This transient block in progression from pro-myelination to myelination, in *Taz*-cKO/*Yap*-cHET, is additional evidence that YAP/TAZ directly regulate SC differentiation and myelination (independently of axon sorting).

## YAP/TAZ are required for proper proliferation of immature Schwann cells

Timely axon sorting requires generation of sufficient SCs (*Benninger et al., 2007*; *Grove et al., 2007*). We therefore analyzed SC proliferation and apoptosis in sciatic nerves of developing *Yap/Taz* cDKO mice. From E17.5 to P60, numbers of Sox10+ SCs in cDKO were reduced to half or less of those in control mice (*Figure 4A*, *Figure 4—figure supplement 1A and B*), and Ki67+ proliferating SCs were substantially reduced at P0 (*Figure 4B*, *Figure 4—figure supplement 1B*). Proliferation was reduced in WT SCs at P4, as promyelinating SCs exited the cell cycle and began to myelinate, and was maintained at similar levels in P4 cDKO SCs, presumably due to residual YAP/TAZ in some cDKO SCs.

To test this possibility and to investigate further the regulation of SC proliferation by YAP/TAZ, we pulse-labeled control and *Yap/Taz* cDKO mice with EdU at E17.5 and P4 to selectively label dividing SCs in S phase. At E17.5, when SC proliferation peaks with active axon sorting, only ~10% SCs were dividing in *Yap/Taz* cDKO, compared to ~30% SCs in control mice (*Figure 4C and E*), concomitant with the reduced SC numbers in cDKO sciatic nerves (*Figure 4A*). At P4, about the same number of SCs were EdU+, and therefore in S phase, in control and cDKO (*Figure 4C*); however, there was weak YAP/TAZ immunoreactivity (ie., residual Yap/Taz) in most of these cDKO EdU+ SCs (for example, *Figure 4F*, #1, #2). We found that less than 1% of SCs completely lacking YAP/TAZ were EdU+, whereas ~15% of SCs with residual YAP/TAZ were dividing (*Figure 4D and F*). Taken together with the recent finding that YAP regulates S-phase entry (*Cabochette et al., 2015*; *Shen and Stanger, 2015*), this result suggests that entrance of SCs into S-phase might be markedly restricted without YAP/TAZ. Collectively, these results show that YAP/TAZ are required for proper proliferation of immature SCs, and that the subsequent reduction in SCs resulted in the failure of timely axon sorting in cDKO.

Laminin receptors, such as integrin $\alpha6\beta1$ and dystroglycan, regulate radial sorting, in part by controlling SC number and proliferation (*Pellegatta et al., 2013*; *Berti et al., 2011*; *Yu et al., 2005*). An earlier analysis attributed defective radial sorting in *Yap/Taz* cDKO largely to downregulation of laminin receptors (*Poitelon et al., 2016*). We therefore examined expression of laminin receptors in our cDKO mice, particularly at P60, and focused on integrin $\alpha6$ because of its obvious downregulation (*Poitelon et al., 2016*). Our immunohistochemical analysis revealed no integrin $\alpha6$ in SCs in P60 cDKO (*Figure 4—figure supplement 2A*). Western blotting and RT-qPCR analysis of cDKO sciatic nerves also showed markedly reduced protein and mRNA expression of integrin $\alpha6$ (*Figure 4—figure supplement 2B and C*). This result therefore confirms the earlier report of Poitelon et al. that YAP/TAZ regulate the expression of integrin $\alpha6$ and presumably other laminin receptors. This result also suggests that radial sorting proceeds, albeit abnormally and remarkably slowly, in the absence of laminin receptors.

Failure of generated SCs to survive could also account for the reduced number in *Yap/Taz* cDKO. Electron microscopy, caspase-3 and TUNEL assays of sciatic nerves taken from cDKO at P4, P18 and P60 did not reveal apoptotic SCs (data not shown). In addition, intramuscular axons and axon terminals showed no axonal degeneration and we observed no denervated muscle fibers at P55 in cDKO (n > 6 muscles, >400 NMJs, *Figure 4—figure supplement 3*). All pre-terminal axons were associated with SC processes (*Figure 4—figure supplement 3B″*, inset). Peri-synaptic terminal SCs, a type of non-myelinating SC, appeared normal and were tightly associated with axon terminals. The axon terminals of P55 *Yap/Taz* cDKO mice frequently extended nerve terminal and SC sprouts, and formed satellite extrasynaptic contacts (*Figure 4—figure supplement 3B′*, arrows), a normal response of axon terminals to muscle paralysis (*Son and Thompson, 1995*)

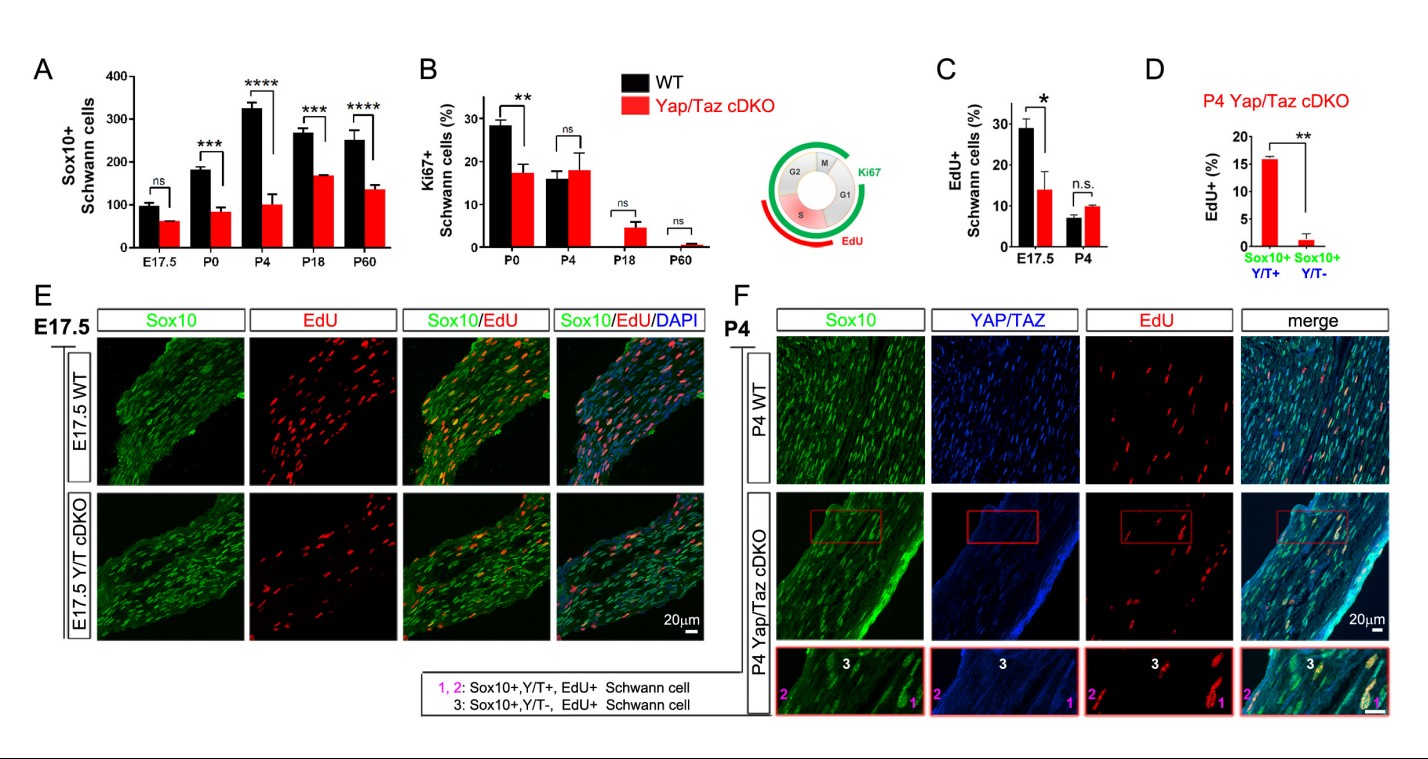

**Figure 4.** YAP/TAZ are required for proper proliferation of immature SCs. (A) Quantitation of total SCs (Sox10+ nuclei) per transverse sciatic nerve section in WT and *Yap/Taz* cDKO mice. n = 3 mice per genotype. ***p<0.001, ****p<0.0001, n.s. = non-significant, 2-way ANOVA with Sidak's multiple comparison test. (B) Percentage of proliferating SCs as determined by co-staining for Ki67 and Sox10. n = 3 mice per genotype. **p<0.01, n.s. = non-significant, 2-way ANOVA with Sidak's multiple comparison test. (C) Quantitation of proliferating SCs in longitudinal sciatic nerve sections at E17.5 and P4, as determined by co-staining for EdU and Sox10. n = 3 (E17.5) or 2 (P4) mice per genotype. *p=0.0192, n.s. = non-significant, 2-way ANOVA with Sidak's multiple comparison test. (D) Quantification of proliferation of SCs in cDKO sciatic nerves at P4; SC nuclei were separately binned as containing detectable YAP/TAZ (Sox10+; Y/T+) or no detectable YAP/TAZ (Sox10+; Y/T-), before counting the % of each subset that was EdU+. n = 2 mice per genotype. **p=0.0044, unpaired Student's t-test. (E, F) Identification of proliferating SCs in longitudinal WT and cDKO sciatic nerve sections. SC nuclei are marked by Sox10 (green), proliferating cell nuclei by EdU incorporation (red), and total cell nuclei by DAPI (blue). (F, bottom insert panel) Identification of proliferating YAP/TAZ+ and YAP/TAZ- SCs at P4, showing examples of EdU+ (proliferating) SCs with no (#3) or weak (#1,#2) YAP/TAZ immunoreactivity. The following figure supplements are available for *Figure 4*.

The following figure supplements are available for figure 4:

**Figure supplement 1.** Schwann cell proliferation during postnatal development.

**Figure supplement 2.** YAP/TAZ are required for *integrin α6* expression in Schwann cells.

**Figure supplement 3.** Neuromuscular junctions of *Yap/Taz* cDKO mice.

## YAP/TAZ are required for Krox20 upregulation

Our findings thus far indicate that SCs lacking YAP/TAZ enter S-phase with difficulty and fail to proliferate in a timely fashion, markedly delaying radial sorting in *Yap/Taz* cDKO mice. These SCs gradually, albeit abnormally slowly, progress through axon sorting, but arrest at the promyelinating stage and fail to initiate myelination. Sequential expression of two core transcription factors is critical for promyelinating SCs to differentiate into mSCs: Oct6 (also known as Scip, Pou3f1) upregulates Krox20/Egr2 in promyelinating SCs and Krox20 upregulates myelin genes and feeds back to turn off Oct6 (*Ghislain and Charnay, 2006*). To investigate the mechanism by which SCs lacking YAP/TAZ arrest as promyelinating SCs, we examined expression of Oct6 and Krox20 in developing and adult cDKO SCs (*Figure 5A–D*). In control nerves, the number of Oct6+ SCs peaked at P4, and then sharply declined. In striking contrast, an abnormally high number of cDKO SCs expressed Oct6 from

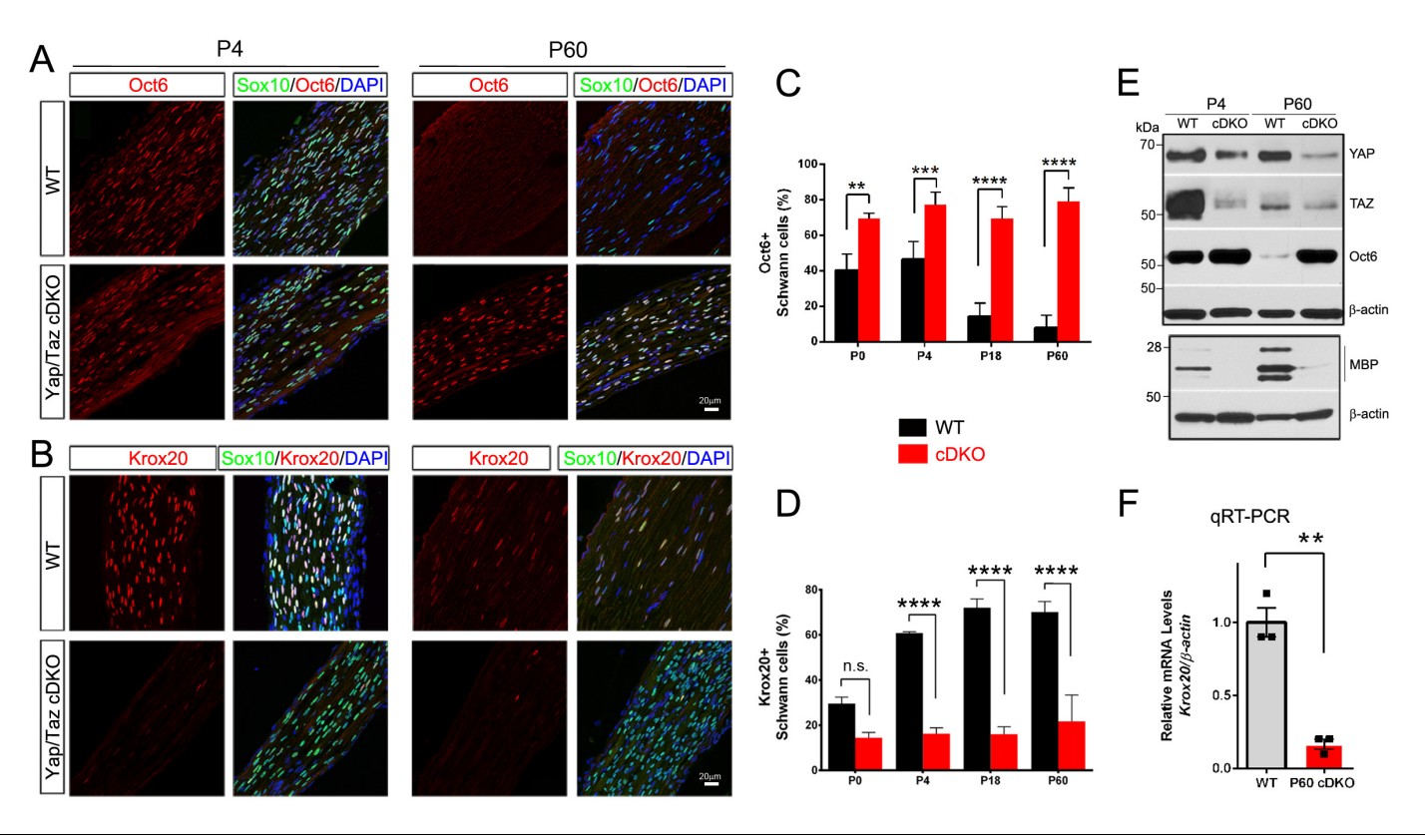

**Figure 5.** YAP/TAZ are required for Krox20 upregulation in differentiating Schwann cells. (**A**) Identification of Oct6+ SC nuclei at P4 and P60 in WT and *Yap/Taz* cDKO, as determined by co-staining for Oct6 (red), Sox10 (SC nuclei; green) and DAPI (all cell nuclei; blue). (**B**) Identification of Krox20+ SC nuclei at P4 and P60 in WT and *Yap/Taz* cDKO, as determined by co-staining for Krox20 (red), Sox10 (SC nuclei, green) and DAPI (all cell nuclei). (**C**) Quantification of Oct6 expression in SC nuclei. n = 4 mice per genotype (WT and cDKO P4) or n = 3 mice per genotype (WT and cDKO P0, (**P18 and P60**). **p<0.01 (**P0**), ***p<0.001, ****p<0.0001, 2-way ANOVA with Sidak's multiple comparison test. (**D**) Quantification of Krox20 expression in SC nuclei. n = 3 mice per genotype (WT P0, WT P4, mutant P18) or n = 2 mice per genotype (mutant P0, mutant P4, WT P18, WT and mutant P60). ****p<0.0001, n.s. = non-significant, 2-way ANOVA, with Sidak's multiple comparison test. (**E**) Western blotting of P4 and P60 WT and *Yap/Taz* cDKO sciatic nerve lysates, using the indicated antibodies and anti-βactin as a loading control. n = 3 experiments. (**F**) Quantitative RT-PCR using *Krox20* and *β-actin*-specific primers and total RNA isolated from WT and *Yap/Taz* cDKO P60 sciatic nerves. Expression of *Krox20* is normalized to that of *β-actin* as an internal control, and WT expression is arbitrarily given the value 1. n = 3 mice per genotype. **p<0.01, unpaired Student's t-test.

P0 to adulthood (*Figure 5A and C*). Notably, cDKO SCs failed to upregulate Krox20: in control mice, ~30% of SCs were Krox20+ at P0, increasing to ~70% from P18 onwards, whereas in cDKO, ~15% of SCs were Krox20+ throughout development and adulthood (*Figure 5B and D*). Western blotting of P4 and P60 cDKO sciatic nerves confirmed abnormal upregulation of Oct6, in parallel with concomitant reduction in YAP/TAZ expression (*Figure 5E*). As expected, Krox20 mRNA expression was also greatly reduced (*Figure 5F*). Together, these results suggest that YAP/TAZ initiate SC differentiation and myelination, presumably by upregulating Krox20 expression in promyelinating SCs.

## YAP/TAZ are required for myelin maintenance: myelinopathy and animal death

To investigate the regulation of adult myelination by YAP/TAZ, we used a tamoxifen inducible *Sox10-creERT2* driver line to inactivate *Yap/Taz* in SCs of adult mice. *Yap/Taz* inducible knockout mice (*Sox10-creERT2; Yap*^fl/fl^; *Taz*^fl/fl^, hereafter *Yap/Taz* iDKO) were indistinguishable from control mice prior to tamoxifen administration. Recombination was induced at 2–3 months of age; efficient SC-selective cre expression was confirmed using a *TdTomato* reporter allele (data not shown) and

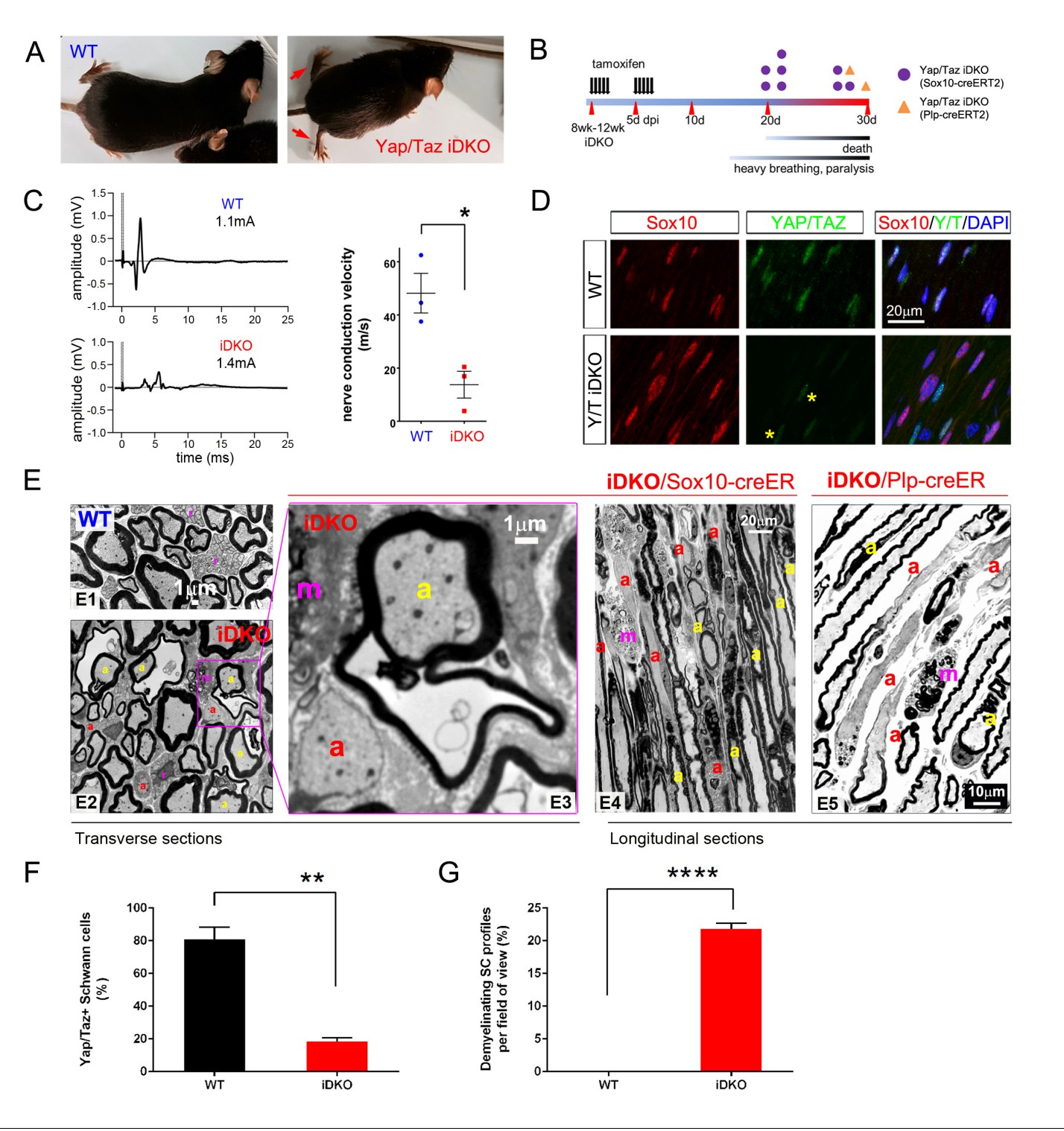

**Figure 6.** YAP/TAZ are required for myelin maintenance. (**A**) Living eight week-old WT and *Yap/Taz* iDKO (*Sox10-Cre-ERT2*) mice 20 days post-first tamoxifen injection. Arrows indicate abnormal splayed gait. (**B**) Cartoon showing timeline of tamoxifen injection and time of sacrifice/ death of iDKO mice due to severity of symptoms. Purple dots: *Sox10-creERT2; Yap*fl/fl; *Taz*fl/fl; Orange dots: *Plp1-creERT2; Yap*fl/fl; *Taz*fl/fl iDKO mice. (**C**) Representative images of CMAPs generated in WT and iDKO mice. Right panel: Nerve conduction velocity, n = 3 mice per genotype. *p=0.0186, unpaired Student's t-test. (**D**) Longitudinal cryosections of sciatic nerves of 11 week-old WT and iDKO (*Sox10-Cre-ERT2*) mice, 20 days after first tamoxifen injection, showing loss of YAP/TAZ (green) in iDKO but not WT SC nuclei, marked by Sox10 (red). All cell nuclei are marked by DAPI staining (blue). Asterisks mark lack of deletion of YAP/TAZ in non-SCs. n = 2 mice per genotype; two sections per mouse. (**E**) Transverse sciatic nerve (E1-3) and longitudinal

*Figure 6 continued on next page*

*Figure 6 continued*

ventral root (**E4, E5**) sections from 11 week old WT, iDKO (*Sox10-Cre-ERT2*; **E2–E4**) and iDKO (*Plp1-Cre-ERT2*; **E5**) mice, 20 days after first tamoxifen injection. (**E1–E3**) TEM of WT (**E1**) and iDKO (**E2, E3**) sciatic nerves. Axons with abnormal myelin profiles are marked with a yellow 'a'; completely demyelinated axons are marked with a red 'a'; myelin-laden macrophages are marked with a red 'm'. n = 3 mice of each genotype. (**E4–E5**) Semi-thin ventral root sections, showing loss of myelin and loosened myelin sheaths in iDKO (*Sox10-Cre-ERT2*) and iDKO (*Plp1-Cre-ERT2*) mice. Demyelinated axons are marked by 'a' and myelin-filled macrophages are marked by 'm'. Note the demyelinated internodes (marked by sets of 'a') contiguous with normally myelinated internodes. (**F**) Bar graph showing percentage of SCs immunopositive for YAP/TAZ 20 days after first tamoxifen injection, in WT and iDKO (*Sox10-Cre-ERT2*) sciatic nerve. (**G**) Quantification of demyelinating SC profiles 20 days after first tamoxifen injection, in transverse sections of WT and iDKO (*Sox10-Cre-ERT2*) sciatic nerve. n = 3 mice per genotype, ****p<0.0001, unpaired Student's t-test. The following figure supplements are available for *Figure 6*.

The following figure supplement is available for figure 6:

**Figure supplement 1.** YAP and TAZ are not expressed in mature oligodendrocytes.

YAP/TAZ immunostaining in SCs (*Figure 6D and F*). Within two weeks of the first tamoxifen injection, *Yap/Taz* iDKO, but not control, mice displayed persistent shivering, ataxia, hunched posture, weight loss and rapid shallow breathing, which progressively worsened until death within the next two weeks (*Figure 6A and B*, *Video 2*, n > 8). To exclude *cre*-driver dependent effects on phenotypes including mouse death, we also generated an inducible *Yap/Taz* iDKO with a *Plp1-creERT2* driver line, which had been used in numerous earlier studies of SC myelination (*Leone et al., 2003*). We found that adult *Plp1-creERT2; Yap*^fl/fl^*; Taz*^fl/fl^ mice exhibited a similar phenotype (eg., *Figure 6E*) and died with respiratory failure within one month after the first tamoxifen injection. Electrical stimulation of iDKO sciatic nerves (*Sox10-creERT2; Yap*^fl/fl^*; Taz*^fl/fl^) between 2 to 3 weeks after the first tamoxifen injection elicited CMAPs with low and dispersed potentials. Nerve conduction velocity (NCV) in iDKO was ~16 m/s, which was markedly reduced from ~50 m/s in control mice (*Figure 6C*).

Electron microscopy of iDKO sciatic nerves revealed numerous demyelinating axons (*Figure 6E2 and G*): some were completely demyelinated (*Figure 6E2 and E3*; red 'a'), others were undergoing demyelination and showed loosened or disorganized myelin sheaths (*Figure 6E2 and E3*; yellow 'a'). Frequent myelin-laden macrophages were additional evidence of ongoing demyelination (*Figure 6E*; 'm'). Longitudinal sections demonstrated an additional striking features of iDKO axons: segmental demyelination with adjacent myelinated or demyelinated internodes (*Figure 6E4*; demyelinated internodes marked by 'a'). Non-myelinating SCs or Remak bundles were unaffected (data not shown), consistent with the absence of nuclear YAP/TAZ in non-myelinating SCs. These results show that YAP/TAZ are required for maintaining SC myelination in adult peripheral nerves.

Histological analysis of *Plp1-creERT2* driven iDKO mice showed similar features of demyelination to Sox10-creERT2 iDKO (*Figure 6*: compare E4 to E5). We also examined the spinal cords and optic nerves of both *Plp1-creERT2-* and *Sox10-creERT2* driven iDKO mice. Oligodendrocytes in adult WT mice did not express YAP/TAZ (*Figure 6—figure supplement 1*), and myelination in the optic nerves in both iDKO mice appeared normal (data not shown). Therefore, central demyelination cannot account for the iDKO phenotypes.

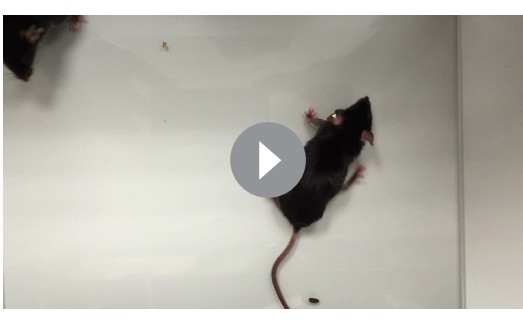

**Video 2.** A movie showing an adult *Yap/Taz* iDKO and a control littermate mouse. The iDKO mouse was recorded at 20 days after the first tamoxifen administration.

## YAP/TAZ are required for mature Schwann cells to express Krox20

Next, we asked whether YAP/TAZ are required for Krox20 expression, as Krox20 is required for myelin maintenance (*Decker et al., 2006*). If so, then administration of tamoxifen should cause deletion of *Yap/Taz* and concomitant reduction of Krox20 in adult iDKO SCs. To test this

possibility, we employed two separate assays because simultaneous immunostaining of Sox10, YAP/ TAZ and Krox20 was not feasible. Notably, immunostaining of iDKO nerves revealed that many DAPI + nuclei had only faint Krox20 immunoreactivity and, only occasional nuclei exhibited strong Krox20 immunoreactivity (*Figure 7A*, arrow). Moreover, YAP/TAZ were virtually undetectable in nuclei that stained weakly for Krox20 (*Figure 7A*, arrowheads). Lastly, Sox10+ SC nuclei in iDKO mice were frequently labeled faintly or not at all for Krox20 (*Figure 7B*, arrowheads; *Figure 7C*). These results suggest that YAP/TAZ maintain adult myelination of peripheral nerves, presumably by promoting Krox20 expression.

## YAP/TAZ-TEAD1 upregulate Krox20 transcription in myelinating developing nerves

Next, we investigated whether YAP/TAZ regulate Krox20 transcription. During developmental myelination, transcription of *Krox20* is mediated by its myelinating Schwann cell enhancer (MSE),~35 kb downstream of its coding region, which binds the transcriptional factors (TFs) Sox10, Oct6, NFAT3/4 and YY1 (*Ghislain and Charnay, 2006*; *He et al., 2007*; *Kao et al., 2009*; *Reiprich et al., 2010*). Because YAP/TAZ lack DNA binding domains, they act on DNA binding-TFs, most notably the TEAD family (*Zhao et al., 2008*). Interestingly, we identified four potential TEAD binding sites in the MSE (*Figure 8A*, *Figure 8—figure supplement 1*) (*Diepenbruck et al., 2014*). We therefore examined the possibility that YAP/TAZ may drive *Krox20* transcription by activating TEADs or other known, MSE-binding TFs. To this end, we performed IP and ChIP-qPCR using P12-P15 and P70 sciatic nerves for developmental and adult myelination, respectively.

We first used western blots followed by immunoprecipitation with anti-YAP/TAZ antibody to study expression of these TFs in P12-P15 sciatic nerves (*Figure 8B*). Oct6 was undetectable, as expected (data not shown; *Jessen and Mirsky, 2005*; *Svaren and Meijer, 2008*). Sox10, YY1, NFAT3, NFAT4 and TEADs 1 and 4 were robustly expressed (*Figure 8B and C1*). Notably, only TEAD1 was detectable in YAP/TAZ immunoprecipitates (*Figure 8B*). The direct association of TEAD1 particularly with TAZ was confirmed by counter immunoprecipitation with anti-TEAD1 antibody (*Figure 8C1*). To investigate whether TEAD3 could be associated with YAP/TAZ, we immunoprecipitated with anti-TEAD1 and anti-TEAD3 and doubled the amount of the input. TEAD1 was present robustly in the immunoprecipitates, but TEAD3 was undetectable (*Figure 8C2*). Therefore, TEAD1 is a partner TF of TAZ for developmental myelination.

We then investigated binding of TEAD1 to the Krox20 MSE. We designed 6 MSE primer sets encompassing the potential TEAD1 binding sites (*Figure 8A*, *Figure 8—figure supplement 1*; E1–E6), and used them to carry out TEAD1 ChIP-qPCR on P12-15 sciatic nerves. To increase the specificity of our assay, we sonicated nerve lysates to produce chromatin fragments of 300–400 bp (data not shown). MSE regions encompassing primer sets E3, E4, and E5 showed strong association with TEAD1 (*Figure 8D*; >50 fold more binding than control). This region included two non-canonical TEAD binding sites and abundant binding sites for Sox10, Oct6, NFAT3/4 and YY1 (*Figure 8A*, *Figure 8—figure supplement 1*). These results suggest that YAP/TAZ-TEAD1 binds to *Krox20* MSE, in order to upregulate *Krox20* for SC differentiation and myelination, and are consistent with an earlier report (*Lopez-Anido et al., 2016*).

## YAP/TAZ-TEAD1 continues to promote Krox20 transcription for myelin maintenance

Krox20 is also required for myelin maintenance (*Decker et al., 2006*), but whether the MSE is obligatory for Krox20 transcription in adult mSCs remains unknown (*Glenn and Talbot, 2013*). We therefore investigated whether TEAD1 is associated with the MSE in adult SCs. We sonicated P70 sciatic nerve lysates to generate chromatin fragments of 300–400 bp, and immunoprecipitated with anti-TEAD1 or control Rabbit IgG antibody. Purified chromatin was used for qPCR; primers were for regions E3, E4 and E5, as these regions were strongly positive for TEAD1 binding at P12–P15 (*Figure 8D*). In addition, we used a primer set encompassing a potential TEAD binding site in the Krox20 promoter, reasoning that TEAD1 may bind to the Krox20 promoter even if the MSE were inactive in adult SCs. Notably, TEAD1 specifically bound to regions encompassing E3, E4 and E5 (>50 fold enriched compared to control IgG), as well as the promoter region (>10 fold enriched). Similarly to P12, the E4 region bound strongly to TEAD1. In contrast to P12-P15, however, binding

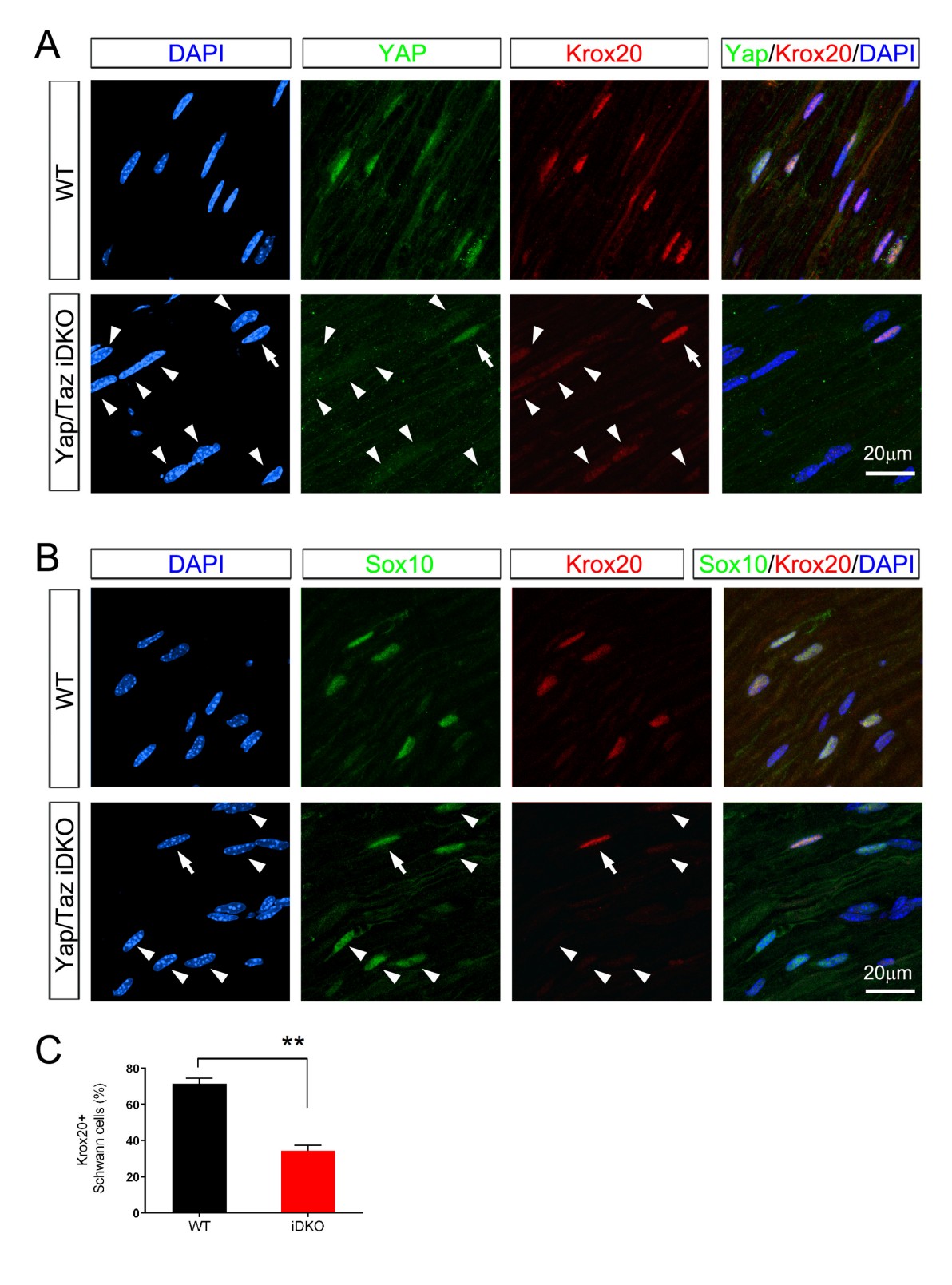

**Figure 7.** Inducible ablation of *Yap/Taz* in adult Schwann cells leads to rapid loss of Krox20. (**A, B**) Immunostaining of longitudinal cryosections of sciatic nerves of WT and iDKO (*Sox10-Cre-ERT2*) mice 20 days after first tamoxifen injection. (**A**) Occasional nuclei, marked with DAPI (blue), exhibit YAP/ TAZ (green) and bright Krox20 (red) immunoreactivity (arrow), while most other nuclei show no YAP/TAZ and faint or no Krox20 immunoreactivity (arrowheads). (**B**) In iDKO sciatic nerves, occasional SC nuclei (Sox10+; green) contain bright Krox20 (red) immunoreactivity (arrow), but most SC nuclei

*Figure 7 continued on next page*

*Figure 7 continued*

exhibit faint or no Krox20 immunoreactivity (arrowheads). (C) Quantification of loss of Krox20 immunoreactivity in Schwann cell (Sox10+) nuclei 20 days after first tamoxifen injection. Both bright and faint immunoreactivity were counted as Krox20+. n = 3 mice per genotype, **p<0.01, Student's unpaired t-test.

to the E5 region was much reduced (~20% of E4, compared to 100% of E4 at P12), suggesting differential regulation of the MSE in adult SCs (*Figure 8E*). These results are the first to demonstrate that the MSE is active in adult mSCs, and suggest that YAP/TAZ-TEAD1 continues to act on the MSE to drive *Krox20* transcription for adult myelination.

Lastly, we investigated TEAD1 expression in SCs at various stages. TEAD1 was strongly expressed by proliferating, differentiating, and mature SCs (*Figure 8F and G*; P4 and P18 data not shown), suggesting that YAP/TAZ-TEAD1 drives SC proliferation, differentiation and maintenance. Non-myelinating SCs (data not shown) and cDKO SCs lacking YAP/TAZ (*Figure 8H*) also expressed TEAD1, suggesting that it is the presence or absence of YAP/TAZ which determines transcriptional regulation of Krox20 by TEAD1.

## Discussion

In this report we identify YAP/TAZ as essential regulators of SC proliferation and myelination. YAP/TAZ are required for 1) SC proliferation important for timely axon sorting, 2) SC differentiation critical for myelination, and 3) maintenance of differentiated myelinating SCs essential for myelin maintenance. Notably, transcriptionally active, nuclear YAP/TAZ have not been detected in fully differentiated cells in adult tissues, and nuclear exclusion (that is, cytoplasmic retention) of YAP/TAZ is widely regarded as requisite for maintaining homeostasis in differentiated cells and tissues (*Piccolo et al., 2014*; *Varelas, 2014*). It was therefore surprising to find that YAP/TAZ are selectively expressed in nuclei of myelinating SCs, and that eliminating them in adulthood evokes demyelination and is rapidly fatal. Our findings are the first to show that YAP/TAZ can be required for adult tissue homeostasis and suggest caution in targeting them for cancer therapy.

We found that proliferation of SCs lacking both YAP/TAZ was markedly inhibited due to repressed entry into S-phase. To match the number of SCs with that of large axons, radial sorting requires a sustained and timely increase in SC proliferation between E16.5 and ~P3 (*Benninger et al., 2007*; *Grove et al., 2007*; *Webster et al., 1973*). *Yap/Taz* cDKO, however, contained only 1/3 of the normal number of SCs at P4 (*Figure 4A*). This number gradually increased but remained substantially below normal. These data therefore strongly suggest that the markedly delayed radial sorting in cDKO mice was primarily due to the failure of SCs lacking YAP/TAZ to generate sufficient SCs at the appropriate time.

SCs lacking both YAP and TAZ were completely arrested at the promyelination stage, like SCs lacking Krox20 (*Decker et al., 2006*; *Topilko et al., 1994*). Furthermore, mRNA and protein levels of Krox20 were concomitantly reduced in SCs of *Yap/Taz* cDKO mice. These results suggest that YAP/TAZ directly regulate SC myelination primarily by activating transcription of *Krox20*. In addition, in our ChIP protocol, we increased specificity by preparing chromatin fragments ~300 bp. in length. This protocol allowed us to pinpoint binding of TEAD1 to specific Krox20 MSE regions. It also allowed us to show much stronger binding of TEAD1 to the MSE in developing SCs (>50 fold increase compared to control) than that recently reported in P15 rat sciatic nerves (*Lopez-Anido et al., 2016*). Our preliminary data also showed that YAP/TAZ bind to the same region of the MSE as TEAD1 (data not shown). Our evidence therefore strongly suggests that a YAP/TAZ-TEAD1 complex initiates SC differentiation and myelination by directly binding to the *Krox20* MSE.

Perhaps the most exciting finding of the present study is that YAP/TAZ-TEAD1 mediate *Krox20* transcription selectively in myelinating SCs. The mechanisms by which adult SCs maintain *Krox20* transcription and thus myelination remain unclear (*Salzer, 2015*). It has been suggested that, once developmental myelination is completed, the *Krox20* MSE becomes inactive, and regulation of TFs that maintain myelination, including Krox20, becomes cell-autonomous, presumably via a feedforward cascade that does not depend on extra- and intracellular signals (*Ghislain et al., 2002*; *Glenn and Talbot, 2013*; *Salzer, 2015*). Our findings, however, strongly imply that signals in adult

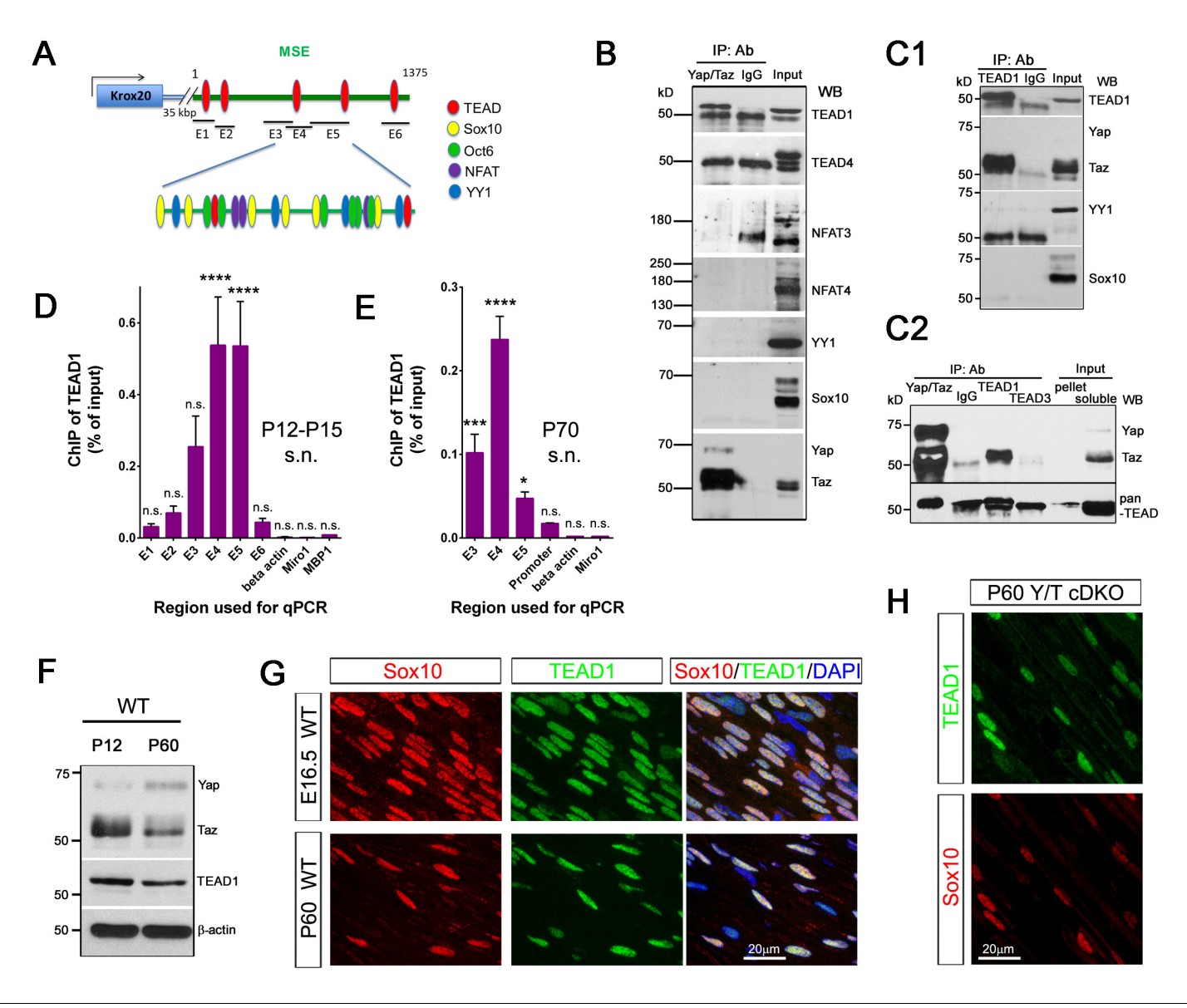

**Figure 8.** YAP/TAZ-TEAD1 regulate *Krox20* transcription during developmental myelination and adult myelin maintenance. (**A**) Schematic showing potential TEAD binding sites in the *Krox20* MSE, in relation to published binding sites for known MSE-binding transcription factors. The regions of the MSE covered by primer sets E1-E6 used for ChIP-qPCR are indicated. (**B, C1, C2**) Immunoprecipitation of P12-P15 WT sciatic nerve lysates with anti-YAP/TAZ (**B**), anti-TEAD1 (**C1**), anti-YAP/TAZ, anti-TEAD1 and anti-TEAD3 (**C2**), plus control Rb IgG (**B, C1, C2**). Immunoprecipitates were blotted with the indicated antibodies. In (**C2**), upper blot was stripped and reprobed with anti-pan-TEAD (lower blot): the band in the Rb IgG lane, migrating similarly to TEAD proteins, is non-specific. n = 3 (**B**) or n = 2 (**C1, C2**) experiments. (**D, E**) ChIP-qPCR of P12-P15 sciatic nerves (**D**) or P70 sciatic nerves (**E**), using anti-TEAD1 antibody or control Rb IgG. Graphs show the amount of qPCR product obtained with the indicated MSE-specific primer sets or negative control primer sets after TEAD1 ChIP. Amounts are given as % of qPCR product obtained with the same primers using input chromatin. Amounts obtained with the same primer sets after control Rb IgG ChIP were all less than 0.003% of input, and are not shown. (**D**) n = 5 (**E1, E2, E4, E6**), n = 4 ((**E5**), n = 3 (*β-actin*), n = 2 (**E3**) independent ChIP assays. (**E**) n = 2 independent ChIP assays. P-values are given for TEAD1 ChIP versus Rb IgG ChIP for each primer set. ****p<0.0001, ***p=0.0001, *p=0.0498, n.s. = non-significant, 2- way ANOVA with Sidak's multiple comparison test. Bar represents mean ± SEM. (**F**) Western blotting of sciatic nerve lysates from WT of P12 or P60 mice, showing persistent expression of TEAD1 and YAP/TAZ. (**G**) Immunostaining of longitudinal sections of WT sciatic nerves, showing that TEAD1 (green) is present in all nuclei (marked by Sox10, red) of proliferating and differentiated SCs. (**H**) Longitudinal nerve sections of P60 *Yap/Taz* cDKO, showing that TEAD1 (green) is present in all cDKO SC nuclei (marked by Sox10, red). The following figure supplements are available for *Figure 8*.

The following figure supplement is available for figure 8:

*Figure 8 continued on next page*

*Figure 8 continued*

**Figure supplement 1.** Annotated sequence of the *Krox20* Myelin Specific Enhancer.

mSCs regulate nucleocytoplasmic shuttling of YAP/TAZ to maintain activation of *Krox20* via its MSE. The MSE region to which TEAD1 binds is enriched with binding sites for TFs, including Sox10, Oct6, NFAT3/4 and YY1 (*Ghislain and Charnay, 2006*; *He et al., 2007*; *Kao et al., 2009*; *Reiprich et al., 2010*). Although our assays indicated no direct interaction between YAP/TAZ and these factors, it is possible that YAP/TAZ-TEAD1 synergize with them by binding to the Mediator complex (*Galli et al., 2015*; *Vogl et al., 2013*). Notably, these TFs are expressed in oligodendrocytes as well as SCs (*He et al., 2007*; *Hornig et al., 2013*; *Schreiber et al., 1997*), whereas YAP/TAZ and Krox20 are exclusively expressed in myelinating SCs, consistent with a critical requirement for YAP/TAZ in *Krox20* expression. It is also interesting that non-myelinating SCs and *Yap/Taz* cDKO SCs express TEAD1, despite absence of YAP/TAZ. TEADs can repress transcription in the absence of YAP/TAZ (*Koontz et al., 2013*; *Zhang et al., 2014*). It is conceivable, therefore, that TEAD1 actively represses *Krox20* transcription in these SCs, consequently maintaining amyelination.

It is also possible that YAP/TAZ regulate myelination by additional pathways unrelated to Krox20. For example, YAP/TAZ might positively regulate Zeb2 transcription, which was recently shown to promote SC differentiation by inactivating repressors of differentiation (*Quintes et al., 2016*; *Wu et al., 2016*). Against this possibility, however, *Zeb2* is downregulated when SCs initiate myelination and it is absent in adult SCs. Another possibility is that YAP/TAZ may activate the mTORC1 pathway to promote myelination (*Kim et al., 2015*; *Norrmén et al., 2014*). In light of the reports that YAP/TAZ are required for Dicer expression (*Chaulk et al., 2014*; *Mori et al., 2014*) and that Dicer ablation blocks SC differentiation (*Pereira et al., 2010*), it is also plausible that YAP/TAZ regulate myelination, at least in part, by regulating miRNA expression.

*Yap/Taz* iDKO mice died as early as 20 days after the first tamoxifen injection, which was unexpected because inducible deletion of neither *Krox20* (*Decker et al., 2006*) nor *Sox10* in SCs (*Bremer et al., 2011*) was lethal. Respiratory failure was the apparent cause of death, which coincided with progressively more shallow and rapid breathing. One speculative explanation is that YAP/TAZ have other transcriptional targets in iDKO mice and/or a cytoplasmic function in adult SCs (*Piccolo et al., 2014*; *Varelas, 2014*; *Yu et al., 2015*), which, when lost, increases the deficits of the iDKO. Alternatively, inducible ablation of YAP/TAZ in adult SCs might have been more successful than that of Krox20 or Sox10 (*Figure 6F* and, *Decker et al., 2006*, *Bremer et al., 2011*).

Our findings show that YAP/TAZ regulate not only proliferation but also differentiation of SCs, driving radial sorting, developmental myelination and myelin maintenance. Notably, YAP/TAZ are present in both nucleus and cytoplasm throughout the stages observed, suggesting that nuclear localization of YAP/TAZ is both inhibited and promoted and, therefore, both positive and negative regulators tightly control their transcriptional activity. In other cell types, Hippo-kinase pathway and F-actin cytoskeleton strongly regulate transcriptional activity of YAP/TAZ and their nucleocytoplasmic shuttling (*Varelas, 2014*). Cell stretching or stiff ECM promotes F-actin- or actomyosin-mediated nuclear entry of YAP/TAZ (*Dupont, 2016*). Similarly, in SCs in vitro, high ECM stiffness (to levels seen in adult sciatic nerve in vivo) or cell stretching, in combination with high concentrations of laminin, promotes nuclear localization of YAP/TAZ and SC differentiation (*Poitelon et al., 2016*; *Urbanski et al., 2016*). Therefore, actin polymerization or actomyosin contractility could conceivably promote transcriptional activity of YAP/TAZ that leads to SC differentiation and myelination. In contradiction to this idea, inhibition of actomyosin contractility promotes SC differentiation in vitro (*Grove and Brophy, 2014*; *Leitman et al., 2011*). In addition, our results show that YAP/TAZ are localized to the nucleus during radial sorting, when ECM stiffness and laminin concentration are low (*Grove and Brophy, 2014*; *Urbanski et al., 2016*). Therefore it is likely that additional pathways are also involved in YAP/TAZ regulation. An intriguing possibility is that YAP/TAZ are downstream of regulators of developmental myelination, including Neuregulin 1-type III, integrin $\alpha6\beta1$, G-protein coupled receptor Gpr126 and Wnt (*Feltri et al., 2002*; *Monk et al., 2011*; *Pereira et al., 2009*; *Petersen et al., 2015*; *Taveggia et al., 2005*; *Azzolin et al., 2014*). Future efforts to identify the

upstream and downstream mediators of YAP/TAZ are likely to generate important basic scientific insights and clinical applications.

## Materials and methods

### Animals and genotyping

All surgical procedures and animal maintenance complied with the National Institute of Health guidelines regarding the care and use of experimental animals and were approved by the Institutional Animal Care and Use Committee of Temple University, Philadelphia, PA, USA. All mice were of either sex, and were on the C57/Bl6 background. Generation of mice carrying targeted loxP sites in the *Yap* and *Taz* genes and the genotyping of these mice have been described previously (*Xin et al., 2013*) (MGI:5446483 and MGI:5544289). Targeted ablation of *Yap* and *Taz* in Schwann cells of embryonic mice was achieved by crossing *Yap*$^{fl/fl}$, *Taz*$^{fl/fl}$ or *Yap*$^{fl/fl}$; *Taz*$^{fl/fl}$ mice with mice heterozygous for *Yap* and/ or *Taz* and transgenic for *Cre* driven by the *P0* promoter (*Feltri et al., 1999*) (MGI:2450448). Targeted ablation of *Yap* and *Taz* in Schwann cells of adult mice was by doing the above crosses of *Yap*$^{fl/fl}$ and *Taz*$^{fl/fl}$ mice with mice transgenic for the inducible Cre recombinase *Cre-ERT2* under control of either the *Plp1* promoter (*Pereira et al., 2009*) (MGI:2663093) or the *Sox10* promoter. The *Sox10-CreERT2* BAC Tg mice were generated with the homologous recombination method (*Yang et al., 1997*) (MGI:5634390) by Dr. Shin Kang at Temple University. Control mice contained floxed *Yap* and/or *Taz* alleles but were not transgenic for *Cre*. Genotyping of *Plp1-CreERT2* mice used *Plp1-Cre* primers (*Pereira et al., 2009*) and of *Sox10-CreERT2* mice used the following primers: CACCTAGGGTCTGGCATGTG and ATCGCGAACATCTTCAGG.

### Tamoxifen administration

Tamoxifen or 4-hydroxy tamoxifen (4-HT; Sigma, St. Louis, MO) was injected into 6–8 week old iDKO mice. Tamoxifen was prepared and injected intraperitoneally (i.p.) at 0.2 mg/g body weight/d, as described (*Grove and Brophy, 2014*). A 20 mg/ ml stock of 4-hydroxy tamoxifen (4-HT) in 100% ethanol was prepared by sonication, and a 1 mg aliquot mixed 1:5 with sunflower oil. After evaporation of the ethanol by speedvac, 1 mg of 4-HT in sunflower oil was injected i.p. twice daily for three days, followed by once daily for two days. Five days later, mice were injected i.p. once daily for five consecutive days.

### Western blotting and immunoprecipitation

For Western blotting and immunoprecipitation, epineurium and perineurium were removed from sciatic nerves, which were then snap-frozen in liquid nitrogen and stored at −80˚C. For Western blotting, sciatic nerves were lysed in RIPA buffer (25 mM Tris pH 7.5, 150 mM NaCl, 1% Triton X-100, 0.5% sodium deoxycholate, 1 mM EDTA, 0.1% SDS), and for immunoprecipitation, lysis was in co-IP buffer (50 mM Tris pH 8.0, 150 mM NaCl, 0.5% Triton X-100, 10% glycerol, 1 mM EDTA), both containing protease and phosphatase inhibitors. Lysis was for 40 min on ice, followed by microcentrifugation at 14,000 rpm for 20 min. Protein concentration was determined using the BCA assay (Thermo Fisher, Waltham, MA).

Immunoprecipitation was overnight at 4˚C, followed by 1 hr with protein A sepharose (preblocked with 1% BSA), both with constant rotation. Lysates and immunoprecipitated complexes in 1x Laemmli buffer were resolved on 8 or 10% SDS-PAGE gels, transferred to nitrocellulose and blotted with the appropriate antibodies. Primary antibodies were used at the following concentrations for Western blots or (if indicated) immunoprecipitations: rabbit anti-Krox20 (Covance, San Diego, CA; 1:400), rabbit anti-Oct6 (Abcam, United Kingdom, ab126746; 1:1000), mouse anti-Myelin Basic Protein (Covance SMI94; 1:1000), mouse anti-beta actin (SIGMA-Aldrich A5441; 1:5000), rabbit anti-Sox10 (Abcam ab155279; 1:20,000), rabbit anti-TEAD1 (Cell Signaling, Danvers, MA, D9 × 2L, #12292; 1:1000; 1:100 for immunoprecipitation), rabbit anti-TEAD3 (Cell Signaling #13224; 1:1000; 1:100 for immunoprecipitation), rabbit anti-pan-TEAD (Cell Signaling D3F7L, #13295; 1:1000; 1:100 for immunoprecipitation), rabbit anti-YAP/TAZ (Cell Signaling D24E4, #8418; 1:1000), rabbit anti-phospho-YAP (Cell Signaling D9W2I #13008; 1:1000), mouse anti-YAP1 (Abcam ab56701; 1: 400), rabbit anti-NFAT3 (Cell Signaling 23E6, #2183; 1:1000), rabbit anti-NFAT4 (Cell Signaling #4998; 1:1000), rabbit anti-YY1 (Abcam ab12132; 1:500). Control whole rabbit IgG for immunoprecipitations

was from Jackson Laboratories (Bar Harbor, ME, 011-000-003) and was used at 1 μg/ml, which was several-fold excess over all antibodies used for immunoprecipitations. Secondary antibodies for Westerns were HRP-linked sheep anti-mouse IgG and donkey anti-rabbit (both GE Healthcare, United Kingdom; 1:5000) and HRP-linked mouse anti-rabbit, light-chain specific (Jackson ImmunoResearch, West Grove, PA, 211-032-171; 1:20,000). Blots were developed using ECL, ECL Plus (GE Healthcare) or Odyssey (LiCor).

## Immunohistochemistry, confocal and widefield fluorescence microscopy

For immunostaining, sciatic nerves were removed and immediately fixed in 4% paraformaldehyde (Pfa)/phosphate buffer for 1 hr. The method for preparation of 7 μm cryosections, and for immunostaining of longitudinal and transverse sciatic nerve sections was as previously described (*Grove et al., 2007*), except that blocking was in 2% BSA/ PBS rather than fish gelatin. For teased fiber preparation, epineurium was removed from fixed sciatic nerves, before teasing of fibers as described (*Sherman et al., 2012*). For whole mount immunostaining of extensor digitorum longus (EDL) and soleus muscles, mice were perfused with 4% Pfa/ PBS, muscle removed, then post-fixed for 20 mins at room temperature. Processing and immunostaining of muscle fibers was as previously described (*Son and Thompson, 1995*). Axons and nerve terminals were labeled with anti-neurofilament and anti-SV2, respectively, while Schwann cells in muscles were labeled with anti-S100 plus anti-p75, in order to detect both intact and denervated Schwann cells. Acetylcholine receptors in muscle were detected using Alexa 647-conjugated a-bungaratoxin (Molecular Probes, Eugene, OR, B35450).

Schwann cells in sciatic nerve cryosections and teased fibers were identified by immunostaining with anti-Sox10. For comparison of total Schwann cell numbers (that is, total Sox10+ nuclei) in sciatic nerves of different genotypes, transverse sections were taken from proximal-, mid- and distal regions of the tibial nerve (three sections per region; nine total), and average Schwann cell number per cross-section was calculated. For calculation of the fraction of Schwann cells expressing a particular protein, longitudinal sections were used, and the fraction of Sox10+ cells expressing that protein was averaged over three sections.

Primary antibodies were used at the following concentrations for immunostaining of cryosections: guinea pig anti-Sox10 (kind gift from Michael Wegner, University of Erlangen, Bavaria, Germany; 1:1000), rabbit anti-Krox20 (Covance; 1:400), rabbit anti-Oct6 (Abcam ab31766; 1:800), rabbit anti-Ki67 (Novocastra NCL-Ki67p; 1:200), rabbit anti-Neurofilament 200 (SIGMA-Aldrich N4142; 1:500), mouse anti-Neurofilament (Covance SMI 312; 1:1000), mouse anti-SV2 (Developmental Studies Hybridoma Bank, Iowa; 1:10), rabbit anti-S100 (DAKO; 1:600), mouse anti-Myelin Basic Protein (Covance SMI 94; 1:2000), goat anti-p75 (Neuromics, Minneapolis, MN, GT15057; 1:500), rabbit anti-TEAD1 (Cell Signaling D9 $\times$ 2L, #12292; 1:200), rabbit anti-pan-TEAD (Cell Signaling D3F7L, #13295; 1:200), rabbit anti-YAP/TAZ (Cell Signaling D24E4, #8418; 1:200), rabbit anti-phospho-YAP (Cell Signaling D9W2I #13008; 1:200), mouse anti-YAP1 (Abcam ab56701; 1: 200), mouse anti-GFAP (SIGMA-Aldrich G3893; 1:500), rabbit anti-cleaved caspase 3 (Cell Signaling #9661; 1:400), anti-CC1. Conventional fluorescence microscopy was performed using Olympus BX53 and Zeiss Axio Imager two microscopes. Images were captured using Hamamatsu ORCA R2 C10600 or Zeiss Axiocam HRm Rev three digital cameras, respectively, and Metamorph software. For confocal microscopy, we used a Leica SP8 confocal microscope, with either a 40 $\times$ 1.3 NA or 63 $\times$ 1.4 NA objective and Leica proprietary software. Acquired stacks were assembled using the maximum projection tool, and figures were prepared using Adobe Photoshop.

## Electron microscopy, histology and morphometry

Mice were perfused intravascularly with 2.5% glutaraldehyde and 4% paraformaldehyde in 0.1M cacodylate buffer pH 7.4. Sciatic nerves and nerve roots were removed and incubated in the same buffer for 90 min at room temperature then overnight at 4°C, with rotation. Specimens were post-fixed for 1 hr with $OsO_4$, dehydrated then embedded in araldite. Semi-thin (0.5 μm) and ultrathin (70 nm) section were cut using an ultramicrotome, and stained with toluidine blue or uranyl acetate and lead citrate, respectively. Semi-thin sections were examined using bright-field on an Olympus BX53 microscope, and images captured using Hamamatsu ORCA R2 C10600 digital camera and Metamorph software.

For analysis of the number of myelin profiles in control vs mutant sciatic nerves, semi-thin sections from proximal-, mid- and distal regions of the sciatic nerve were taken, and two non- overlapping images of each section were taken using the 60x objective: these images were termed 'field of view' (FOV). The average number of myelin profiles in each FOV was calculated, using ImageJ software. For electron microscopy, stained ultrathin sections were examined using a JEOL 1010 electron microscope fitted with a Hamamatsu digital camera and AMT Advantage image capture software.

## TUNEL and proliferation assays

The percentage of Schwann cells in S-phase at E17.5 and P4 was measured by EdU incorporation. Timed pregnant females and P4 pups were respectively injected intraperitoneally and subcutaneously with 80 µg/ g body weight of EdU dissolved in PBS. Eighty minutes later, E17.5 embryos and P4 pups were sacrificed, and sciatic nerves removed and fixed for 1 hr in 4% Pfa; longitudinal 7 µm cryosections were used for analysis. EdU incorporation was identified using the Click-iT EdU Alexa Fluor 555 labeling kit (Molecular Probes); the recommended protocol was followed, and labeling was for 1 hr. After EdU labeling, sections were immunostained with anti-Sox10 to identify Schwann cells (notably, many proliferating cells in the sciatic nerves were not Schwann cells). The percentage of Schwann cells (Sox10+ nuclei) that were EdU+ was determined by counting of confocal images using ImageJ software. The Promega Deadend TUNEL assay kit was used to investigate Schwann cell apoptosis in longitudinal and transverse 7 µm cryosections. The recommended protocol was followed, with the following modification: after rehydration in PBS, sections were treated with L.A.B. retrieval solution for 5 min (Polysciences, Inc 24310–500), then washed 3x in PBS, prior to permeabilization in 0.2% Triton-X-100 for 10', as per the recommended protocol. As a positive control, sections were treated with 7.5 U/ml DNase1 for 10' prior to the TUNEL assay; as a negative control, TdT enzyme was omitted from the assay. After the TUNEL assay, sections were immunostained with anti-Sox10 to identify Schwann cells, and TUNEL+ Schwann cells in confocal images were counted using ImageJ software.

## Real-time quantitative RT-PCR

Sciatic nerves were removed from P60 control and cDKO mice, and total RNA extracted using an RNeasy Lipid Tissue Mini Kit (Qiagen). RT-qPCR was performed using the Power SYBR Green RNA-to-CT one-step kit (Applied Biosystems), specific primers, plus 20 ng RNA per reaction. *β-actin* was used as an endogenous reference gene, and relative expression of *Krox20* in control and cDKO samples was calculated according to the Comparative Ct method (*Livak and Schmittgen, 2001*), using StepOne Software (V 2.3). Primer sets used were as follows: 5'-AGGCCCCTTTGACCAGATGA-3' and 5'-AAGATGCCCGCACTCACAAT-3' for *Krox20* (*Bremer et al., 2011*); 5'-CTCCCCGAAGTTC TTCCCAT-3' and 5'TAAGGTTGCTGTTACAGACATTG-3' for *ITGA6*; 5'-ACCCTAAGGCCAACCG TGA-3' and 5'-ATGGCGTGAGGGAGAGCATA-3' for *β-actin*; 5'-GACATCTTCTGGTCAAAGATAC TTC-3' and 5'-GCAGGAACGTTCAGTTGCG-3' for *Yap*; 5'-GTCCATCACTTCCACCTCGG-3' and 5'-CGACTTGCTGGTGTTGGTGA-3' for *Taz*. Samples were run in triplicate, and experiments performed three times.

## ChIP assay

ChIP assays were done using eight sciatic nerves from P12-P15 wild-type mice per immunoprecipitation, using a modified version of a published protocol (*Pal et al., 2004*). Sciatic nerves were collected in PBS on ice. The PBS was replaced by PBS/1% formaldehyde containing protease inhibitors, and nerves were rapidly diced using scissors. Nerves were crosslinked for 10' with rotation at room temperature, washed with PBS/0.15M glycine for 5' with rotation at room temperature, then washed 3x with ice-cold PBS. Nerves were lysed using ChIP lysis buffer (50 mM Tris pH 8.1, 100 mM NaCl, 5 mM EDTA, 0.5% SDS plus protease inhibitors) for 30' at 4°C, then pelleted by microfugation at 14K for 15'. Supernatant was removed, and pellets were resuspended in 1 ml of ChIP IP buffer (50 mM Tris pH 8.6, 0.3% SDS, 1.7% Triton X-100, 5 mM EDTA plus protease inhibitors). Chromatin was sheared in an ice water bath by sonication using a Fisher Scientific FB120 sonic dismembrator set at 75% amplitude, with 10 cycles of 10'' on/50'' off. After shearing, chromatin was clarified by micro-centrifugation at 14K for 20 min, snap-frozen in liquid nitrogen and stored at −80°C. 1 ml aliquots of chromatin were pre-cleared using pre-blocked PAS (Millipore 16–157), then 50 µl removed and

stored as 'input' chromatin. The remaining chromatin was immunoprecipitated overnight with 0.4 µg rabbit anti-TEAD1 (Cell Signaling #12292), 4 µg rabbit anti-TAZ (Cell Signaling #4883) and 0.5 µg or 5 µg of whole rabbit IgG, respectively (Jackson Laboratories 011-000-003). Immune complexes were collected using pre-blocked PAS (for 2 hr, then washed x2 sequentially with: low salt buffer (0.1% SDS, 1% Triton X-100, 2 mM EDTA, 150 mM NaCl, 20 mM Tris pH 8.0); high salt buffer (0.1% SDS, 1% Triton X-100, 2 mM EDTA, 500 mM NaCl, 20 mM Tris pH 8.0); LiCl buffer (1% NP40, 1% Na deoxycholate, 1 mM EDTA, 250 mM LiCl, 10 mM Tris pH 8.0); TE (10 mM Tris, 1 mM EDTA, pH 8.0), all containing protease inhibitors. After the final TE wash, chromatin was eluted from the beads 2x for 20' with gentle vortexing in 150 µl Elution buffer (1% SDS, 0.1M NaHC0$_3$) at room temperature. 0.2M NaCl was added, and chromatin de-crosslinked overnight at 65°C. Samples were treated with RNase A (0.3 mg/ml) at 37°C for 1 hr, then proteinase K (0.6 mg/ml) at 55°C for 1 hr, before purification of immunoprecipitated chromatin using QIAquick columns (Qiagen). Chromatin was eluted in 25 µl 10 mM Tris pH 8.0, and stored at −20°C. qPCR reactions were in 10 µl, using Power SYBR green PCR mastermix (Applied Biosystems # 4367659), 4 µl of sample (1:80 dilution of input and 1:20 dilution of IPs) and the following primers: Krox20 MSE: E1 forward: CCTCTCCGGGGTTGGAC TA; E1 reverse: GGGCCACCCACAGTTACCTT; E2 forward: GGCCCTCCATTTATTTCCCTG; E2 reverse: TGCTGGAAACGAAACAACAAACT; E3 forward: TGTGACAAACCCATCTCGCA; E3 reverse: TCAGCTTTGTGAAGGGCTCA; E4 forward: CGGACTTGCATTGCACAGAA; E4 reverse: CCCGTACATCCACTCACACA; E5 forward: CCTTCACAAAGCTGAAATCCTG; E5 reverse: CA TTTCGTCTTTGGGCTCAT; E6 forward: CATGGTTAGGAGGAACAAACTG; E6 reverse: CTATC TCAACCTGCCAATCCA. Negative Controls: Miro1 forward 1: GCAATTTCTTGGGCTGTGATATAG; Miro1 reverse 1: GCTCTTAGAGGCACTTTGGAT; β-actin forward: GGCTGTATTCCCCTCCATCG; β-actin reverse: CGTCCCAGTTGGTAACAATGCC; MBP forward 1: TCTGAGCAAATGCTTCTAAGCTG ; MBP reverse 1: CTAGAGCAGAAAGTGAGCTTCC.

## Electrophysiology

The procedure for measuring nerve conduction velocity was an adaption of a previously used proto-col (*Schulz et al., 2014*). Mice were deeply anaesthetized, and hair completely removed from the hindquarters. A 27g percutaneous needle electrode was inserted subcutaneously over the femur to stimulate the sciatic nerve at two locations: one proximal to the spinal cord, at the hip joint, and the other 1 cm distal to this site. The stimulation reference electrode (26G percutaneous needle) was inserted over the back. Bipolar EMG electrodes were inserted subcutaneously in the digital interosse-ous muscles, to record compound action potentials in response to activation of the sciatic nerve. Stimulation and recording were conducted on both hindlimbs. The muscle response was first tested with single pulses (100 µs duration) to determine the threshold and level of activation producing a stable compound action potential in the muscle. The level of stimulation which produced stable responses was then used for two series of stimulations with 2–20 s long trains of biphasic square wave pulses (100 µs duration) delivered at 1–5 Hz. The nerve was rested for 2 min between runs. Stimulation was delivered via an isolated pulse stimulator (A-M Systems Model 2100, A-M Systems Inc, Carlsborg, WA). EMG compound muscle action potential (CMAP) responses were recorded with a differential AC amplifier (A-M Systems Model 1700; gain 1 k– 10 k; pass band 10–50 KHz). Data was sampled at 50 KHz using LabVIEW (National Instruments Corporation, Austin, TX) running on a personal computer. Multiple recordings were taken at each location to ensure a minimum of 10 qual-ity responses for each location and animal. Electrophysiological data was analyzed using Igor Pro (Wavemetrics Inc, Lake Oswego, OR).

## Behavioral tests

*Foot printing gait analysis*: Mouse paws were dipped in non-toxic water-based paints: forepaws: pink; hindpaws: blue. Mice were then allowed to walk along an 80 cm x 10 cm white paper runway. *Grip strength test:* Mice were allowed to grip the grid of a BIO-GS3 grip strength testing device (Biosep, France), using both fore- and hindpaws, then were gently pulled by the tail, parallel to the grid. The maximum force prior to the release of all of the mouse's paws from the grid was recorded. This was repeated with forepaws alone, and grip strength of hindpaws was calculated by subtraction of the forepaw measurement from the total grip strength measurement. The apparatus was cleaned using alcohol between trials. *Temperature preference/ aversion test:* Two-choice temperature

preference/ aversion assays were used to determine hot and cold thermal thresholds, using methods essentially, as described (*Fisher et al., 2015*). Briefly, mice were placed in a Plexiglass chamber containing a reference plate fixed at 25°C plus an adjacent variable temperature test plate (Bio-T2CT, BioSeb, France). Mice were allowed to habituate to the chamber for 3 min with both plates at 25°C, then the test plate temperature was increased to 49°C or decreased to 10°C in 3°C increments, at 3 min intervals. Movement of mice between reference and test plates was recorded using a video tracking system, and percentage of time at each temperature calculated. Preference/ aversion to cold vs hot temperatures was assayed on separate days.

## Data analysis

In each experiment, data collection and analysis was performed identically, regardless of mouse genotype. Data is presented as mean +/- SEM. Statistical analysis was done using unpaired Student's t-test or 2-way ANOVA with either Sidak's or Tukey's multiple comparison tests, according to the number of samples and the analysis of mice at multiple ages. Sample sizes were similar to those employed in the field and are indicated in main text, methods or figure legends. Data distribution was assumed to be Gaussian, but was not formally tested, and $p < 0.05$ was considered significant.

P values, differences between means (DM), 95% confidence intervals (CI) of differences between mean, t-distributions and degrees of freedom (df) were as follows: *Figure 2 (G)*: unpaired Student's t-test. *Yap/Taz* cDKO myelinating Schwann cells per FOV, p=0.0034, DM = −358.7 + /−54.54, CI = −518.4 to −198.9, t = 6.233, df = 4. *Figure 3C*: 2-way ANOVA, with Sidak's multiple comparison test. *Taz* cKO/*Yap* cHET and *Yap/Taz* cDKO myelinating and promyelinating Schwann cells per FOV, *Taz*-cKO/*Yap*-cHET P18: promyelinating, p=0.0002, DM = −144.0, CI = −208.2 to −79.82, t = 6.306, df = 11; P18 myelinating, p<0.0001, DM = 667.8, CI = 541.3 to 794.4, t = 14.83, df = 11; P60 promyelinating: p>0.9999, DM = 0, CI = −71.76 to+71.76, t = 0, df = 11; P60 myelinating: p=0.2271, DM = 96.25, CI = −45.25 to 237.8, t = 1.912, df = 11. YAP/TAZ-cDKO, P18 promyelinating: p=0.0569, DM = −62.5, CI = −126.7 to 1.68, t = 2.737, df = 11; P18 myelinating: p<0.0001, DM = 778.7, CI = 652.1 to 905.2, t = 17.29, df = 11; P60 promyelinating: p<0.0001, DM = −210.3, CI = −274.5 to −146.2, t = 9.211, df = 11; P60 myelinating: p<0.0001, DM = 377, CI = 250.4 to 503.6, t = 8.372, df = 11. *Figure 4*: (A-C) 2-way ANOVA, with Sidak's multiple comparison test. (A) *Yap/Taz* cDKO total Sox10+ Schwann cells per sciatic nerve, E17.5: p=0.3244, DM = 35.67, CI = −18.57 to 89.9, t = 1.887, df = 18; P0: p=0.0009, DM = 98.95, CI = 38.31 to 159.6, t = 4.682, df = 18; P4: p<0.0001, DM = 224.2, CI = 169.9 to 278.4, t = 11.86, df = 18; P18: p=0.0008, DM = 100.5, CI = 39.86 to 161.1, t = 4.755, df = 18; P60: p<0.0001, DM = 116.2, CI = 61.96 to 170.4, t = 6.147, df = 18. (B) *Yap/Taz* cDKO Ki67+ Schwann cells, P0: p=0.0025, DM = 11, CI = 3.733 to 18.27, t = 4.244, df = 16; P4: p=0.9235, DM = −1.9, CI = −9.167 to 5.367, t = 0.7331, df = 16; P18: p=0.3356, DM = −4.567, CI = −11.83 to 2.7, t = 1.762, df = 16; P60: p=0.9995, DM = −0.5, CI = −7.767 to 6.767, t = 0.1929, df = 16. (C) *Yap/Taz* cDKO EdU+ Schwann cells, E17.5: p=0.0192, DM = 15.03, CI = 3.133 to 26.93, t = 3.738, df = 6; P4: p=0.8322, DM = −2.8, CI = −17.38 to 11.78, t = 0.5684, df = 6. (D) Unpaired t-test. *Yap/Taz* cDKO, YAP/TAZ+ vs YAP/TAZ- EdU+ Schwann cells at P4, p=0.0044, DM = −14.73 + /- 0.9772, CI = −18.93 to −10.53, t = 15.07, df = 2. *Figure 5*: (C, D) 2-way ANOVA, with Sidak's multiple comparison test. (C) *Yap/Taz* cDKO Oct6+ Schwann cells in sciatic nerve, P0: p=0.0027, DM = −29.24, CI = −48.94 to −9.545, t = 4.106, df = 18; P4: p=0.0001, DM = −30.68, CI = −45.93 to −15.42, t = 5.56, df = 18; P18: p<0.0001, DM = −55.26, CI = −88.79 to −53.55, t = 9.272, df = 18; P60: p<0.0001, DM = −71.17, CI = −88.79 to −53.55, t = 11.17, df = 18. (D) *Yap/Taz* cDKO Krox20+ Schwann cells in sciatic nerve, P0: p=0.1146, DM = 15.13, CI = −2.908 to 33.17, t = 2.491, df = 11; P4: p<0.0001, DM = 44.62, CI = 26.58 to 62.65, t = 7.347, df = 11; P18: p<0.0001, DM = 56.06, CI = 38.02 to 74.1, t = 9.231, df = 11; P60: p<0.0001, DM = 48.35, CI = 28.59 to 68.11, t = 7.268, df = 11. (F) Unpaired t-test. *Yap/Taz* cDKO, Krox20 mRNA in Schwann cells at P60. p=0.0213, DM = −0.695 + /- 0.1031, CI = −1.139 to −0.2515, t = 6.742, df = 2. *Figure 6*: (C) Unpaired t-test. Nerve conduction velocity in iDKO mice. p=0.0186, DM = −34.4' ± 8.984, CI = −59.36 to −9.468, t = 3.83, df = 4. (F) Unpaired t-test. YAP/TAZ+ Schwann cells in iDKO mice. p=0.0018, DM = −62.55 + /- 5.872, CI = −81.24 to −43.86, t = 10.65, df = 3. (G) Unpaired t-test. Demyelinating Schwann cells in iDKO mice. p<0.0001, DM = 21.77 + /- 0.9387, CI = 19.16 to 24.37, t = 23.19, df = 4. *Figure 7*: (C) Unpaired t-test. Krox20 + Schwann cells in iDKO mice. p=0.0038, DM = −37.02 + /- 4.497, CI = −51.33 to −22.71, t = 8.232, df = 3. *Figure 8*: (D, E) 2-way ANOVA with Sidak's multiple comparison test. (D) TEAD1

ChIP from P12-P15 sciatic nerves. E1 primers: p=0.9997, DM = 0.0294, CI = −0.1733 to 0.2321, t = 0.408, df = 44; E2 primers: p=0.9498, DM = 0.0684, CI = −0.1343 to 0.2711, t = 0.9492, df = 44; E3 primers: p=0.1994, DM = 0.2535, CI = −0.06706 to 0.5741, t = 2.225, df = 44; E4 primers: p<0.0001, DM = 0.5356, CI = 0.3329 to 0.7383, t = 7.432, df = 44; E5 primers: p<0.0001, DM = 0.5335, CI = 0.3068 to 0.7602, t = 6.622, df = 44; E6 primers: p=0.9966, DM = 0.0428, CI = −0.1599 to 0.2455, t = 0.5939, df = 44; beta actin primers: p>0.9999, DM = 0.001333, CI = −0.2604 to 0.2631, t = 0.01433, df = 44. (E) TEAD1 ChIP from P70 sciatic nerves. E3 primers: p=0.0001, DM = 0.1019, CI = 0.05544 to 0.1484, t = 6.89, df = 12; E4 primers: p<0.0001, DM = 0.234, CI = 0.1875 to 0.2805, t = 15.82, df = 12; E5 primers: p=0.0498, DM = 0.0465, CI = 4.058e-005 to 0.09296, t = 3.144, df = 12; Krox20 promoter primers: p=0.8830, DM = 0.016, CI = −0.03046 to 0.06246, t = 1.082, df = 12; beta actin primers: p>0.9999, DM = 0.0005, CI = −0.04596 to 0.04696, t = 0.03381, df = 12; Miro1 primers: p>0.9999, DM = 0.001, CI = −0.04546 to 0.04746, t = 0.06761, df = 12. *Figure 2—figure supplement 1*: (C) 2-way ANOVA, with Sidak's multiple comparison test. *Yap/Taz* cDKO, percentage of Schwann cells containing residual YAP/TAZ, P4: p=0.0007, DM = 45.7, CI = 28.87 to 62.53, t = 8.559, df = 5; P60: p=0.0003, DM = 61.5, CI = 43.06 to 79.94, t = 10.51, df = 5. (D) Unpaired t-test. *Yap/Taz* cDKO vs WT, relative *Yap* and *Taz* mRNA levels. *Yap*: p=0.0009, DM = −0.8333 + /- 0.09428, CI = −1.095 to −0.5716, t = 8.839, df = 4; *Taz*: p=0.0215, DM = −0.45 + /- 0.1458, CI = −0.8067 to −0.0933, t = 3.087, df = 6. *Figure 2—figure supplement 2*: (B) 2-way ANOVA, with Sidak's multiple comparison test. *Yap/Taz* cDKO fore- and hindlimb grip strength. Forelimb, 10 week: p=0.0028, DM = 47.57, CI = 20.23 to 74.9, t = 4.774, df = 8; 16 week: p=0.0030, DM = 47.01, CI = 19.67 to 74.34, t = 4.718, df = 8. Hindlimb, 10 week: p=0.0105, DM = 96.98, CI = 26.93 to 167.0, t = 3.798, df = 8; 16 week: p=0.0113, DM = 95.68, CI = 25.64 to 165.7, t = 3.747, df = 8. (D) 2-way ANOVA, with Tukey's multiple comparison test. *Yap/Taz* cDKO, mice weights, P60: WT female vs *Yap/Taz* cDKO female, p=0.0005, DM = 4.7, CI = 2.158 to 7.242, q = 7.599, df = 14; WT male vs *Yap/Taz* cDKO male, p<0.0001, DM = 9.75, CI = 6.636 to 12.86, q = 12.87, df = 14. *Figure 2—figure supplement 3*. (E) 2-way ANOVA with Sidak's multiple comparison test. Number of small and large axon bundles containing large axons in *Yap/Taz* cDKO sciatic nerve. Small bundles (2–5 axons) p<0.0001, DM = −86.1, CI = −105.9 to −66.27, t = 11.91, df = 8; Large bundles (>5 axons) p=0.7342, DM = −5.3, CI = −25.13 to 14.53, t = 0.7331, df = 8. *Figure 4—figure supplement 2*: (C) Unpaired Student's t-test. *Integrin α6* mRNA levels in *Yap/Taz* cDKO P60 sciatic nerve. p=0.039, DM = −0.533 + /- 0.1764, CI = −1.023 to −0.04361, t = 3.024, df = 4.

## Acknowledgements

We thank Alan Tessler and members of Son laboratory for critical reading of the earlier versions of the manuscript. We thank Dr. Eric Olson for *Yap* and *Taz* floxed mice, and Drs. Ueli Suter and Kelly Monk for *Plp1-creERT2* mice, and Dr. Michael Wegner for Sox10 antibody. *Plp1-creERT2* mice were generated by Dr. Ueli Suter using patented *Cre-ERT2* construct developed by Dr. Pierre Chambon at GIE-CERBM.

## Additional information

### Funding

| Funder | Grant reference number | Author |
|---|---|---|
| National Institute of Neurological Disorders and Stroke | NS079631 | Young-Jin Son |
| Shriners Hospitals for Children | 86600 | Young-Jin Son |
| National Institute of Neurological Disorders and Stroke | NS076401 | Bassel E Sawaya |
| National Institute of Mental Health | MH093331 | Bassel E Sawaya |
| National Institute of Neurological Disorders and Stroke | NS095070 | Young-Jin Son |

The funders had no role in study design, data collection and interpretation, or the decision to submit the work for publication.

## Author contributions

MG, Conceptualization, Data curation, Formal analysis, Methodology, Writing—original draft, Writing—review and editing; HK, AJK, SBH, JZ, JYC, Data curation, Formal analysis; MS, MH, Formal analysis, Methodology; RP, BES, SHK, Resources, Methodology; SK, S-HC, Conceptualization, Resources, Methodology; MFB, Resources, Formal analysis, Methodology; MAL, Resources, Data curation, Formal analysis; Y-JS, Conceptualization, Formal analysis, Writing—original draft, Writing—review and editing

## Author ORCIDs

Young-Jin Son, http://orcid.org/0000-0001-5725-9775

## Ethics

Animal experimentation: This study was performed in strict accordance with the recommendations in the Guide for the Care and Use of Laboratory Animals of the National Institutes of Health. All of the animals were handled according to approved institutional animal care and use committee (IACUC) protocols (#4254, #4255) of the Temple University.

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
