## [Decision Letter]

Thank you for submitting your article "Yap/Taz directly regulate myelin formation and maintenance by upregulating Krox20" for consideration by *eLife*. Your article has been favorably evaluated by Anna Akhmanova (Senior Editor) and five reviewers, one of whom, Klaus-Armin Nave (Reviewer #1), is a member of our Board of Reviewing Editors. The following individual involved in review of your submission has agreed to reveal their identity: Michael Wehr (Reviewer #5).

The reviewers have discussed the reviews with one another and the Reviewing Editor has drafted this decision to help you prepare a revised submission.

Summary:

Grove et al. study Schwann cell-specific Yap and Taz cKO and DKO mice and show that these Hippo signaling transcriptional coactivators factors are essential regulators for PNS myelination. Overall, this study is important as it further reveals a key function of the Yap/Taz signaling system in Schwann cell differentiation and myelination. The paper extends and partly corrects findings by the Swaren and Feltri labs that reported a block of axonal sorting in essentially the same double mutant mice. The authors identify a strong delay (but not a block) of axonal sorting. Instead the authors make a subsequent failure of Krox20 upregulation in the "promyelinating" SC responsible for the myelination defect. They discover TEAD1 as a critical cofactor of Taz to directly regulate Krox20 transcription. The authors also report the requirement of continued Yap/Taz expression in adult Schwann cells to maintain myelin. The overall technical quality of the data is high, and presentation is clear and convincing. Although the concept of Yap/Taz regulation is not entirely novel, they provide additional Yap/Taz functions in Schwann cell development and myelin maintenance.

Essential revisions:

1) An important point in the paper is the efficient recombination achieved in all generated mutants. Nevertheless, both western blotting analyses and immunohistochemistry in Figure 2—figure supplement 1, have residual Yap/Taz expression in the cDKO. They should perform qPCR analyses for Yap/Taz mRNA to confirm efficient ablation in cDKO.

2) The paper by Poitelon et al. presented convincing data that the axonal sorting defects in the Yap/Taz conditional mutants are primarily caused by downregulation of laminin receptors α6β1 integrin and dystroglycan (with minor differences in proliferation), whereas Grove et al. argue equally convincingly that delayed S phase entry and reduced proliferation are the main reasons. It would be important to know whether Grove et al. also see the downregulation of laminin receptors.

3) Radial sorting impairment is present at all time points analyzed and to corroborate the delay the authors should calculate the number of SC in 1:1 association with axons at all time points analyzed.

4)Subsection “Yap/Taz are required for timely axon sorting”, third paragraph: It is troubling to read that some 20% of DKO SC remain Yap/Taz positive, although "weakly", with this staining remaining constant between P4 and P20 (the data are not shown, except for Figure 4 which shows a lot of residual non-nuclear Yap/Taz). This suggests that even after successful recombination these proteins remain quite stable, possibly due to their physical anchorage in the cells (yet at a different level than the compensatory expression in heterozygous mice). It also brings up the question how early both genes are expressed (i.e. prior to P0-Cre expression) and could different turnover rates of Yap/Taz (possibly related to environmental/nutritional/stress factors) relate to phenotypical differences of DKO mice in different labs.

Related issues raised:

In the third paragraph of the subsection “Yap/Taz are required for myelination of sorted axons”, the authors claim (with reference to Figure 2) that the remaining MBP staining in Yap/Taz cDKO is from "rare SCs which retained strong Yap/Taz immunoreactivity".

Subsection “Yap/Taz are required for myelination of sorted axons”, first paragraph and Figure 2—figure supplement 1: what could be the reason for "faint signals" in 20% of cells at that adult age? Cre – recombination is early and should be all or none.

Also, the IHC of Figure 6 shows SC expressing low levels of Yap/Taz.

5) The authors conclude that the reduction of Krox20 expression (see title) causes the myelination delay, but also report that ~15% of SCs are Krox20+ throughout development and adulthood (subsection “Yap/Taz are required for Krox20 upregulation”). If failing Krox20 expression were the only problem, shouldn't these SC proceed to myelination? Here also the conclusions should be more precise.

6) The fact that tamoxifen-dependent Yap/Taz deletion in the adult by Sox10-CreERT2 or Plp-CreERT2 leads to an astonishingly rapid death is attributed to the observed changes in peripheral myelin and the PNS. The PNS defects are clearly shown, and it is equally obvious that there is no major structural alteration in CNS myelin. Nevertheless, one should be more cautious in claiming that the PNS defects are the cause of death.

Given that Sox10 is expressed in other neural crest-derived cell types, does Yap/Taz deletion in other Sox10-expressing cells in Sox10- creERT2-driven iDKO mice cause acute mouse death? The authors mentioned as data-not-shown that Plp-creERT2-driven iDKO mice gave a similar phenotype in demyelination, it would be important to show the data to confirm the demyelination or axonal loss phenotype.

7) Figure 1—figure supplement 1: why is Yap localized in nuclei of Taz KO samples, unlike the control? This is an important point as validation of these antibodies is crucial for the entire study.

8) The authors claim that at P60 the majority of fibers reach a 1:1 association (subsection “Yap/Taz are required for timely axon sorting”, first paragraph), despite the significant impairments in overall SC number and proliferation. Is there an effect on axon numbers in mutant nerves?

9) Figure 4: cDKO nerves have an important reduction in SCs number throughout development (Figure 4). Surprisingly this reduction is not corroborated by a substantial defect in SCs proliferation. The graph in Figure 4 shows a significant decrease in SC proliferation at P0, but normal or even increased levels at P4, P18 and P60. How do the authors reconcile these results given that there is no effect on apoptosis? The experiments supporting these conclusions, in particular the IHC in Figure 4—figure supplement 1 at P18 do not show colocalization between Sox10 and Ki67.

10) In Figure 4 the staining for Yap/Taz is not clear. It seems that all SC are Yap/Taz positive, but this is in sharp contrast with Figure 2—figure supplement 1, where only 60% of cells are Yap/Taz+.

11) In Figure 6, Yap/Taz iDKO mice exhibit a strong slowing of nerve conduction from ~50 m/s to ~15 m/s, which would suggest the loss of many large diameter axons. However, this phenomenon does not match the morphological features of the mutant nerves where many large diameter axons remain present. The authors should provide quantification data showing the degree of demyelination, axonal loss and reduction of Krox20 in SCs to reflect the remarkable decrease in conduction velocity. Is it possible that the phenotype in Figure 6 is due to a reduced numberof SC as in development? Have the authors checked for SC numbers, survival and proliferation in iDKO?

12) The authors show that Tead1 binds the MSE element of the Krox20 gene by ChIP-PCR, and claim that Yap/Taz interaction of Tead1 regulates Krox20 gene expression. However, Tead1 transcription factor can bind to many elements and its binding may not necessarily alter Krox20 gene expression. The Yap/Taz regulation of Krox20 may not even be direct. The authors should mutate the putative Tead1 binding sites to demonstrate that Krox20 transcription is inhibited. It would be also important to know whether Yap/Taz interaction with Tead1 is required for activating Krox20 expression and/or the MSE element activity.

13) The authors do not present any mechanisms of upstream Hippo signalling, independent or dependent on Lats and Mst kinases, mechanical forces, involvement of actin cytoskeleton, etc.. Therefore, the authors should at least discuss these aspects and reference relevant work.

[Editors' note: further revisions were requested prior to acceptance, as described below.]

Thank you for resubmitting your work entitled "Yap/Taz initiate and maintain Schwann cell myelination by upregulating Krox20" for further consideration at *eLife*. Your article has been favorably evaluated by Anna Akhmanova (Senior Editor) and five reviewers, one of whom is a member of our Board of Reviewing Editors.

The authors have carefully addressed all comments and questions that were raised by the referees. We acknowledge that this was a relatively long list that was contributed by several experts in the field, who now had a chance to study the revised manuscript. Before *eLife* can accept the paper, the authors should address a few remaining points that mainly require rewording of text.

1) While it has been clearly stated in the rebuttal letter that the main discrepancy between the present study and that of Poitelon et al. is due to a more thorough analysis of YAP/TAZ in adulthood, this is less evident in the manuscript itself (Results section, subsection “Yap/Taz are required for timely axon sorting”, third paragraph). Please adjust.

2) We are not convinced that there is "better axon sorting in distal nerves of cDKO at P18" (subsection “Yap/Taz are required for timely axon sorting”, third paragraph), a conclusion based merely on low magnification semi-thin sections. Please be more cautious.

3) The correlation of "weak staining in YAP/TAZ" and the failure of mutant SC to enter S phase is too speculative. Unless the expression of cyclins is provided, this conclusion should be softened (subsection “Yap/Taz are required for S-phase entry and proper proliferation of immature Schwann cells”, second paragraph).

4) There is no strong evidence of TAZ/YAP directly activating Krox20 in Schwann cells and the authors themselves stated "These results suggest that Yap/Taz directly regulate Schwann cell myelination most likely by activating transcription of Krox20" (Discussion, third paragraph). Thus, the paper's title ("Yap/Taz initiate and maintain Schwann cell myelination by upregulating Krox20") appears too strong of a statement. YAP/TAZ may regulate many other factors required for Schwann cell myelination. In the absence of evidence demonstrating the ability of TAZ/YAP to upregulate Krox20, e.g. by overexpression studies, the title should be adjusted.

5) The authors added "new data" for the iDKO/PlpCreERT phenotype as a single image in Figure 6E5. However, the same image was shown in the previous manuscript (Figure 6E3 in the first submission), where it was labeled as iDKO/Sox10-CreERT. Please check and correct.

6) The authors had been asked to provide data for SC numbers, survival and proliferation in iDKO. In their response, the authors claim to have performed these experiments but do not provide data ("Lastly, as suggested by the reviewer, we have carried out a small number of additional experiments to check SC numbers and proliferation in iDKO: We have found that SCs are not reduced and that proliferation appears only marginally above background. We plan to provide these data when we report the results of our ongoing study."). For a convincing statement, the authors should at least provide a figure for the referees.

---

## [Author Response]

*Essential revisions:*

*1) An important point in the paper is the efficient recombination achieved in all generated mutants. Nevertheless, both western blotting analyses and immunohistochemistry in Figure 2—figure supplement 1, have residual Yap/Taz expression in the cDKO. They should perform qPCR analyses for Yap/Taz mRNA to confirm efficient ablation in cDKO.*

We observed that, in addition to myelinating SCs, fibroblasts, perineurial cells and presumably also vascular cells strongly express Yap/Taz both in developing and mature sciatic nerves (e.g., asterisks in Figure 2—figure supplement 1, Figure 6). Therefore, we primarily used immunohistochemistry with a SC specific Sox10 antibody to evaluate SC-selective deletion of Yap/Taz in cDKO mice (see also our response to #4). Nonetheless, as suggested, we have carried out RT-qPCR for Yap/Taz mRNA. Total RNA was extracted from P60 sciatic nerves of 3 WT and 3 cDKO mice and used for RT-qPCR with primers in exons 2 and 3 of Yap or exons 3 and 4 of Taz, because exon 3 is deleted in both Yap and Taz knockout mice (Xin et al., 2011). We detected ~22% of Yap mRNA and ~48% of Taz mRNA in the cDKO, compared to WT. These results are now provided as Figure 2—figure supplement 1. These results are comparable to those we obtained with Western blotting of Yap and Taz at P60 (see Figure 5). The reason for the apparently small decrease in Taz mRNA at P60 is not inefficient recombination; Taz recombination is efficient, as evident from the large decrease in Taz protein levels at P4 in cDKO compared to WT (Figure 5). Please note that there is residual background Taz protein at P4 in cDKO, which is maintained at similar levels at P60 in cDKO. Please also note that at P60, WT Taz protein levels are much reduced compared to P4, and are only ~2-fold higher than background; hence it is not surprising that at P60, WT Taz mRNA is similarly only ~2-fold over background. We attribute the background levels of Yap mRNA, Taz mRNA and protein to a combination of perineurial cells, fibroblasts, vascular cells and residual partially recombined SCs (i.e., <20% SCs; see our response to #4).

*2) The paper by Poitelon et al. presented convincing data that the axonal sorting defects in the Yap/Taz conditional mutants are primarily caused by downregulation of laminin receptors α6β1 integrin and dystroglycan (with minor differences in proliferation), whereas Grove et al. argue equally convincingly that delayed S phase entry and reduced proliferation are the main reasons. It would be important to know whether Grove et al. also see the downregulation of laminin receptors.*

We thank the reviewer for this important point. First, it is important to note that Poitelon et al. analyzed mainly Taz-cKO/Yap-cHET mice, whereas we used Yap/Taz cDKO mice. Although Poitelon et al. reported no reduction in total SCs in Taz-cKO/Yap-cHET (despite the much reduced SC proliferation at P3 that they observed), they commented that “…by P20 the total number of SCs was decreased in double-cKO mice, which were not used in subsequent studies…”. Thus, Poitelon et al. also observed a reduction in number of SCs in Yap/Taz cDKO mice.

Second, as suggested, we have now examined laminin receptor expression in Yap/Taz cDKO sciatic nerves (Figure 4—figure supplement 2). We concentrated on integrin α6, which appeared most strongly downregulated in Taz-cKO/Yap-cHET, whereas downregulation of dystroglycan or other laminin receptors seemed less obvious (Poitelon et al., 2016). We used immunohistochemistry, Western blotting, and RT-qPCR to show that integrin α6 is indeed strongly downregulated in Yap/Taz cDKO sciatic nerves at P4 and P60; this new data appears in Figure 4—figure supplement 2. Residual integrin α6 observed in cDKO in Westerns and RT-qPCR is likely due to strong expression of integrin α6 in perineurial cells (asterisks in Figure 4—figure supplement 2, and Poitelon et al., 2016). Notably, in the P60 cDKO, no integrin α6 is associated with single axons or with axons in small bundles (Figure 4—figure supplement 2); i.e., radial sorting is able to proceed (abnormally slowly) in the absence of integrin α6 in Yap/Taz cDKO SCs.

Lastly, earlier studies showed that integrin α6β1 and dystroglycan synergize to increase SC number and proliferation, thereby regulating radial sorting, and that ablation of laminin prevents radial sorting due to reduced SC number and proliferation (Pellegatta et al., 2013, Berti et al., 2011, Yu et al., 2005). It is therefore possible that downregulation of laminin receptors in Yap/Taz cDKO SCs *contributes* to reducing SC proliferation, thereby *contributing* to delaying, but not preventing, radial sorting.

*3) Radial sorting impairment is present at all time points analyzed and to corroborate the delay the authors should calculate the number of SC in 1:1 association with axons at all time points analyzed.*

As suggested, we have calculated the number of SCs in 1:1 association with axons at P18 and P60. We calculated the number of 1:1s per field of view in 3 WT, 3 Taz-cKO/Yap-cHET and 3 Yap/Taz cDKO mice; we counted the number of promyelinating- and the number of myelinating 1:1s at each time point. The data is newly presented in Figure 3. This shows that at P18, there is a significant delay at the promyelination stage in Taz-cKO/Yap-cHET sciatic nerves, as we described in the original submission. Furthermore, while WT and Taz-cKO/Yap-cHET sciatic nerves have no promyelinating SCs at P60, Yap/Taz cDKO sciatic nerves have a significant number, ~210 promyelinating SCs per FOV. Hence, Yap/Taz cDKO SCs gradually (abnormally slowly) sort large-caliber axons, but there is a block in progression from promyelination to myelination.

*4) Subsection “Yap/Taz are required for timely axon sorting”, third paragraph: It is troubling to read that some 20% of DKO SC remain Yap/Taz positive, although "weakly", with this staining remaining constant between P4 and P20 (the data are not shown, except for Figure 4 which shows a lot of residual non-nuclear Yap/Taz). This suggests that even after successful recombination these proteins remain quite stable, possibly due to their physical anchorage in the cells (yet at a different level than the compensatory expression in heterozygous mice). It also brings up the question how early both genes are expressed (i.e. prior to P0-Cre expression) and could different turnover rates of Yap/Taz (possibly related to environmental/nutritional/stress factors) relate to phenotypical differences of DKO mice in different labs.*

*Related issues raised:*

*In the third paragraph of the subsection “Yap/Taz are required for myelination of sorted axons”, the authors claim (with reference to Figure 2) that the remaining MBP staining in Yap/Taz cDKO is from "rare SCs which retained strong Yap/Taz immunoreactivity".*

*Subsection “Yap/Taz are required for myelination of sorted axons”, first paragraph and Figure 2—figure supplement 1: what could be the reason for "faint signals" in 20% of cells at that adult age? Cre – recombination is early and should be all or none.*

*Also, the IHC of Figure 6 shows SC expressing low levels of Yap/Taz.*

We understand the concerns of the reviewer regarding the ~20% SCs that we counted as Yap/Taz positive in Yap/Taz cDKO. We have carried out additional analyses to confirm efficient recombination/deletion in our Yap/Taz cDKO and iDKO mice. These include new qRT-PCR (Figure 2—figure supplement 1, Figure 5), counting axon bundles (Figure 2—figure supplement 3), counting SC profiles (Figure 3), additional immunostaining (Figure 2E2, 4F), additional Western blotting (Figure 5), counting Yap/Taz+, Krox20+ SCs and SC profiles in iDKO (Figure 6, Figure 7). We also strengthen our statements in the text about the reasons for excluding the possibility that the delayed radial sorting that we report is due to incomplete deletion of Yap/Taz in our cDKO mice.

First, Figure 4, to which the reviewer refers, was adjusted for greater overall brightness with the intention of making these small magnification images more visible. In addition, the buffer in the Click-It kit that we used to identify EdU+ nuclei seemed to increase background in subsequent immunostaining with other antibodies. We have now replaced these images with images in which brightness has not been adjusted. Yap/Taz immunoreactivity in both the nucleus and cytoplasm of cDKO SCs is either completely undetectable (e.g., #3 SC) or low (e.g., #1, 2) compared to WT SCs. Please also note that the newly provided Figure 2E2 shows two types of SCs that we counted as Yap/Taz positive in cDKO: #1 SC, which retains “strong Yap/Taz immunoreactivity” and is also MBP+; and #2 SC, which is MBP-negative and only weakly immunopositive for Yap/Taz. Because we also observed many more SCs (e.g., #3, 4, 5, 6 SCs) that exhibit absolutely no Yap/Taz immunoreactivity in cDKO, we counted #2 type SCs as Yap/Taz-positive although they exhibit very low or “faint signals”. These weakly labeled SCs represent most of the ~20% SCs that we considered to have “residual Yap/Taz” in our rigorous quantitative analysis.

We are not certain, however, of the specificity of this residual Yap/Taz immunoreactivity and we speculate that the weak immunoreactivity at least in part may be non-specific background labeling of some SC nuclei. We have screened more commercial antibodies but cannot find additional antibodies that specifically recognize both Yap and Taz (e.g., Figure 1—figure supplement 1). Please also note that, because non-SCs (presumably perineurial cells and fibroblasts) strongly express Yap/Taz both in developing and mature sciatic nerves, we primarily rely on SC-selective immunohistochemistry, rather than Western blotting or qRT-PCR to evaluate recombination/deletion in our cDKO and iDKO mouse lines. Another speculative explanation, which we have now described better in the result section, is that most of the residual Yap/Taz immunoreactivity in ~20% SCs may represent an undeleted allele of Yap, which we found to be insufficient to complete myelination (Figure 3 Taz-cKO/Yap-cHET).

Another possibility, which the reviewer raised, is that these “residual Yap/Taz” are proteins that remain stably in the SC nuclei, even after extended periods. Yap/Taz are expressed in Sox10+ neural crest derived cells at E10.5 (Serinagaoglu et al., 2015); hence Yap/Taz are indeed expressed prior to when P0-Cre drives SC-selective deletion of Yap/Taz. However, we think it unlikely that residual Yap/Taz would remain in cDKO SCs even at P60. This is based on our finding that most adult SCs lack Yap/Taz 3 weeks after inducible ablation; i.e., nuclear Yap/Taz does not appear to be exceptionally stable in adult Schwann cells.

We agree with the reviewer that it is important to consider whether different turnover rates of Yap/Taz could explain phenotypic differences between cDKO mice in our study and that of Poitelon et al., but we do not think that there is a phenotypic difference. Poitelon et al. examined cDKO mice up to P20, and most of their data was from Taz-cKO/Yap-cHET mice. The phenotypes of our and Poitelon et al.’s Yap-cKO, Taz-cKO, Yap-cKO/Taz-cHET and Taz-cKO/Yap-cHET mice are all quite similar up to ~P20. For example, Poitelon et al. reported that cDKO mice had a complete block in radial sorting at P20. Similarly, we observed a major block in radial sorting at P18 in the proximal sciatic nerves. Notably, however, we found that more sorting had occurred at P18 in distal nerves. Poitelon et al. did not examine cDKO mice after P20, whereas we examined them up to P60, when we observed more advanced radial sorting.

Lastly, regarding Figure 6, we now provide new quantification data (Figure 6) to show that our iDKO mice lines effectively induced deletion of Yap and Taz.

*5) The authors conclude that the reduction of Krox20 expression (see title) causes the myelination delay, but also report that ~15% of SCs are Krox20+ throughout development and adulthood (subsection “Yap/Taz are required for Krox20 upregulation”). If failing Krox20 expression were the only problem, shouldn't these SC proceed to myelination? Here also the conclusions should be more precise.*

Please note that the percentage of Krox20+ SCs (~15%) is similar to that of Yap/Taz+ SCs (~20%) in Yap/Taz-cDKO sciatic nerves. Moreover, similar to our procedure for counting of Yap/Taz+ SCs in the cDKO, we included those SCs that were only weakly immunopositive as Krox20+ SCs. Figure 5 shows that few SCs have high levels of Krox20, and that most have lower levels or do not express it. Notably, Le et al., (Le et al., 2005) have generated hypomorphic Krox20 mice (Egr2lo/lo) that express mRNA and protein levels of Krox20 that are reduced 3-fold compared to WT sciatic nerves. SCs in these Egr2lo/lo mice are arrested at the promyelination stage, similar to Krox20-KO SCs. Therefore, we would not anticipate that Yap/Taz cDKO SCs expressing low levels of Krox20 would proceed to myelination. As suggested, we have modified conclusive statements.

*6) The fact that tamoxifen-dependent Yap/Taz deletion in the adult by Sox10-CreERT2 or Plp-CreERT2 leads to an astonishingly rapid death is attributed to the observed changes in peripheral myelin and the PNS. The PNS defects are clearly shown, and it is equally obvious that there is no major structural alteration in CNS myelin. Nevertheless, one should be more cautious in claiming that the PNS defects are the cause of death.*

*Given that Sox10 is expressed in other neural crest-derived cell types, does Yap/Taz deletion in other Sox10-expressing cells in Sox10- creERT2-driven iDKO mice cause acute mouse death? The authors mentioned as data-not-shown that Plp-creERT2-driven iDKO mice gave a similar phenotype in demyelination, it would be important to show the data to confirm the demyelination or axonal loss phenotype.*

We agree with the reviewer that the causes of Yap/Taz iDKO mouse death should be interpreted cautiously and analyzed further. We have amended related statements. Our response to #11 comment offers one possible explanation (see below). We were also surprised that both lines of Yap/Taz iDKO mice driven by Sox10-creERT2 or Plp-creERT2 died within a month after the 1^st^ tamoxifen injection, exhibiting similar segmental demyelination phenotypes in the peripheral nerve. We are currently examining whether induced deletion of Yap/Taz in SCs has more detrimental effects besides segmental demyelination and whether it acutely paralyzes neuromuscular functions. We also plan to examine other cell types potentially affected in both lines of iDKO mice. As suggested, we now provide a semi-thin image of Plp-creERT2 driven iDKO sciatic nerve (Figure 6E5), which shows a demyelination phenotype similar to that in Sox10-creER iDKO nerves (Figure 6E1-E4).

*7) Figure 1—figure supplement 1: why is Yap localized in nuclei of Taz KO samples, unlike the control? This is an important point as validation of these antibodies is crucial for the entire study.*

Yap is localized in the nuclei of both control and Taz cKO sciatic nerves. Arrows in the figure now mark examples of SC nuclei in the control. It is also noteworthy that Yap is present in both the cytoplasm and nucleus, but not in all nuclei, in the control. Other figures show that Yap is expressed in myelinating, but not in non-myelinating, SCs (Figure 1).

*8) The authors claim that at P60 the majority of fibers reach a 1:1 association (subsection “Yap/Taz are required for timely axon sorting”, first paragraph), despite the significant impairments in overall SC number and proliferation. Is there an effect on axon numbers in mutant nerves?*

We regret the confusing statements. We did not mean that radial sorting was almost complete in cDKO at P60 and now corrected this misconception in the Results section. We also provide new data consisting of counts of axon bundles of unsorted axons and promyelinating SCs in P60 cDKO (Figure 2—figure supplement 3, Figure 3). Although we observed many sorted axons in Yap/Taz cDKO at P60, we frequently observed small bundles of unsorted axons (e.g., Figure 2—figure supplement 3, Figure 3) and occasionally large axon bundles (e.g., Figure 2—figure supplement 3; low magnification image of ventral root). More specifically, at P60, while there are no bundles of unsorted axons in WT, cDKO sciatic nerves contain 86.1 small bundles (2-5 axons; ****P<0.0001) and 5.3 large axon bundles (>5 axons) per field of view (Figure 2—figure supplement 3). This compares to 210 promyelinating SCs per field of view in the cDKO at P60 (Figure 3). Therefore, a considerable number of axons remained unsorted in cDKO mice at P60 (Figure 2—figure supplement 3), consistent with their markedly reduced number of SCs (Figure 4).

There are several reasons why we did not count axon numbers in P60 cDKO nerve, for which extensive quantitative EM analysis would be required for accurate counting of amyelinated atrophied axons, including unsorted axons tightly packed in small bundles. First, our finding that a considerable number of axons remained unsorted precludes having to postulate that substantial axon death occurred in cDKO nerves. Second, our EM analysis of cDKO nerves revealed no signs of axon degeneration. This result is consistent with earlier studies of developmental amyelination that reported no axon loss. Third, our comprehensive analysis of muscle innervation in P55 cDKO mice did not show degeneration of intramuscular axons or axon terminals; all muscle fibers remained innervated by single axons, exhibiting no abnormal patterns of innervation or denervation (n>6 muscles, >400 NMJs, Figure 4—figure supplement 3). In summary, we are quite confident suggesting that axons in cDKO do not degenerate at least up to P60.

*9) Figure 4: cDKO nerves have an important reduction in SCs number throughout development (Figure 4). Surprisingly this reduction is not corroborated by a substantial defect in SCs proliferation. The graph in Figure 4 shows a significant decrease in SC proliferation at P0, but normal or even increased levels at P4, P18 and P60. How do the authors reconcile these results given that there is no effect on apoptosis? The experiments supporting these conclusions, in particular the IHC in Figure 4—figure supplement 1 at P18 do not show colocalization between Sox10 and Ki67.*

Because non-SCs, in addition to Sox10+ SCs, proliferate, there are Ki67+ but Sox10- cells in P4 WT, P4 cDKO and P18 cDKO. These proliferating non-SCs might have given the erroneous impression to the reviewer that there is no Sox10/ Ki67 colocalization in the figure. We have now specifically marked Sox10+ and Ki67+ cells in the Figure 4. The unique strength of our studies of Yap/Taz cDKO and iDKO mice is in our use of SC-specific Sox10 antibody, which enabled us to exclude the responses of non-SCs in our analyses.

In WT sciatic nerves, SC proliferation peaks between E16.5 and P0, and then decreases as radial sorting comes to an end and SCs initiate myelination (Jessen and Mirsky, Nature Rev. Neurosci. 2005). Our data show markedly impaired SC proliferation in cDKO during the peak period (E17.5 and P0; Figure 4). Consistent with this finding, we observed markedly reduced numbers of SCs in P4 cDKO mice (Figure 4). We believe that the “…normal or even increased levels of SC proliferation” in P4 cDKO is largely due to residual Yap/Taz in cDKO SC nuclei (Figure 4). The “…normal or even increased levels of proliferation” in P18 and P60 cDKO is highly likely because radial sorting continues abnormally and SCs fail to differentiate (i.e., to initiate myelination) in Yap/Taz cDKO mice. It is important to note, however, that, although cDKO SCs continue to proliferate, their proliferation sharply decreases to insignificant levels by P18. Consistent with this result, we observed only a slight increase in SC number in cDKO after P4 (Figure 4). Therefore, we consider cDKO SC proliferation after P4 to be only minimal and think that there is no discrepancy in SC proliferation and number in our analysis of cDKO mice, which showed no SC apoptosis at least up to P60.

*10) In Figure 4 the staining for Yap/Taz is not clear. It seems that all SC are Yap/Taz positive, but this is in sharp contrast with Figure 2—figure supplement 1, where only 60% of cells are Yap/Taz+.*

In addition to myelinating SCs, we frequently observed non-SCs (presumably fibroblasts, perineurial and vascular cells) that strongly express Yap/Taz both in developing and mature sciatic nerves (e.g., asterisks in Figure 2—figure supplement 1, Figure 6). These Yap/Taz+ non-SCs often give the mistaken impression, particularly at low magnifications such as in Figure 4, that all SCs are Yap/Taz positive. In addition, the buffer in the Click-It kit that we used to identify EdU+ nuclei seemed to increase background in subsequent immunostaining, which might have caused some background-stained SC nuclei to appear Yap/Taz positive. We have now replaced Figure 4 with an image that better displays SCs not expressing Yap/Taz (i.e., non-myelinating SCs) even at low magnifications. We used high magnifications and a SC specific, Sox10 antibody, to count Yap/Taz+ SCs in Figure 2—figure supplement 1. The Click-It kit was not used in this analysis.

*11) In Figure 6, Yap/Taz iDKO mice exhibit a strong slowing of nerve conduction from ~50 m/s to ~15 m/s, which would suggest the loss of many large diameter axons. However, this phenomenon does not match the morphological features of the mutant nerves where many large diameter axons remain present. The authors should provide quantification data showing the degree of demyelination, axonal loss and reduction of Krox20 in SCs to reflect the remarkable decrease in conduction velocity. Is it possible that the phenotype in Figure 6 is due to a reduced number of SC as in development? Have the authors checked for SC numbers, survival and proliferation in iDKO?*

We respectfully disagree with the reviewer that demyelination alone cannot explain markedly decreased nerve conduction velocity and that it is necessary to postulate additional massive loss of large-caliber axons. It is our understanding that demyelination of an axon *at the level of one internode* may even cause conduction block along the axon: “In paranodal demyelination, the initial outward current at the next node is dissipated over a larger area. If the next node can be depolarized to threshold after a larger duration current, saltatory conduction is slowed but preserved; with more severe demyelination insufficient current is available to depolarize the next node to threshold and the action potential will extinguish” (Franssen and van den Bergh, 2006). In addition, earlier studies have commonly reported slow components (<5 m/s) of dispersed nerve conduction, as we observed in iDKO mice, and attributed them to unmyelinated large-caliber axons (e.g., Bremer et al., 2011).

As suggested, we have examined more semithin and EM sections of both lines of iDKO mice at 20 days after 1^st^ tamoxifen injection, and observed no evidence of substantial axon loss (data not shown, see below). We have also quantified Yap/Taz+ SCs (Figure 6), demyelinating profiles (Figure 6) and Krox20+ SCs (Figure 7), as requested by the reviewer. Notably, we found that ~22% of axons were demyelinating or demyelinated on analyses of transverse sections (i.e., at the level of one internode). Although demyelination seems only partial (i.e., ~22%), axons that appear normal at one internode may have multiple demyelinated internodes at other locations along the entire length of sciatic nerve. This notion is strongly supported by our images of *longitudinal* sections of both lines of iDKO mice, wherein along a single axon, demyelinated internodes are adjacent to myelinated internodes (e.g., demyelinated internodes marked by red ‘a’ in Figure 6E4, 6E5). Therefore, it is quite possible, in our opinion, that segmental (not uniform) demyelination that is widespread among the axons of iDKO mice can markedly reduce nerve conduction and be fatal.

Nonetheless, as mentioned above in comment #6, we do not exclude the possibility that induced ablation of Yap/Taz not only causes demyelination but also affects myelinating SCs in additional harmful ways. This is the focus of our ongoing study, which we hope will soon provide additional insights into the robust phenotype and acute death of Yap/Taz iDKO mice. Lastly, as suggested by the reviewer, we have carried out a small number of additional experiments to check SC numbers and proliferation in iDKO: We have found that SCs are not reduced and that proliferation appears only marginally above background. We plan to provide these data when we report the results of our ongoing study.

*12) The authors show that Tead1 binds the MSE element of the Krox20 gene by ChIP-PCR, and claim that Yap/Taz interaction of Tead1 regulates Krox20 gene expression. However, Tead1 transcription factor can bind to many elements and its binding may not necessarily alter Krox20 gene expression. The Yap/Taz regulation of Krox20 may not even be direct. The authors should mutate the putative Tead1 binding sites to demonstrate that Krox20 transcription is inhibited. It would be also important to know whether Yap/Taz interaction with Tead1 is required for activating Krox20 expression and/or the MSE element activity.*

The fact that TEAD1 binds to the MSE in vivo makes it highly unlikely that TEAD1 may have no role in regulating Krox20 expression. TEADs bind two families of transcriptional cofactors: Yap/Taz and Vgll proteins; interestingly, Yap/Taz and Vgll compete for binding to TEAD, but regulate different sets of genes (Pobbati et al., 2012); hence, as we found that Yap/Taz are required for Krox20 expression, it suggests that Yap/Taz complex with TEAD1 to regulate Krox20. In support of this, Svaren’s group recently reported that TEAD1 siRNA led to downregulation of Krox20 expression in primary rat SCs (Lopez-Anido et al., 2016). Also, Tricaud’s group reported that co-expressing Yap and TEAD1 constructs induced maximal Krox20 promoter activity in primary rat SCs (Fernando et al., 2016), strongly suggesting that Yap directly regulates Krox20 transcription, through TEAD1. Notably, our ChIP analysis also shows that TEAD1 binds to the Krox20 promoter 10-fold over control levels, and our preliminary ChIP data shows Yap/Taz binding to the same region of the Krox20 MSE as TEAD1. These data therefore provide additional support for the notion that TEAD1 mediates Krox20 upregulation by Yap/Taz.

We completely agree with the reviewer, however, that TEAD1 is highly unlikely to be the sole transcriptional factor with which Yap/Taz interacts, and that more direct evidence is required to establish Yap/Taz-TEAD1 regulation of Krox20. We have begun comprehensive in vitro and in vivo studies to learn how Yap/Taz regulates Krox20 transcription, which will include mutating putative TEAD1 binding sites in the Krox20 MSE and promoter and use of TEAD binding mutants of Yap/Taz, as the reviewer suggested. In addition, we plan in vivo pharmacological and genetic approaches targeting TEAD1 (Liu-Chittenden et al., 2012). We appreciate the reviewer’s valuable comments. We have now toned down our statements related to the claim.

*13) The authors do not present any mechanisms of upstream Hippo signalling, independent or dependent on Lats and Mst kinases, mechanical forces, involvement of actin cytoskeleton, etc.. Therefore, the authors should at least discuss these aspects and reference relevant work.*

As requested, we now provide the following paragraph as a discussion of potential upstream signals regulating Yap/Taz in SCs.

“Our findings show that Yap/Taz regulate not only proliferation but also differentiation of SCs, driving radial sorting, developmental myelination and myelin maintenance. […] Therefore, it is likely that additional pathways are also involved in Yap/Taz regulation. An intriguing possibility is that Yap/Taz are downstream of regulators of developmental myelination, including Neuregulin 1-type III, integrin α6β1, G-protein coupled receptor Gpr126 and Wnt (Feltri et al., 2002, Monk et al., 2011, Pereira et al., 2009, Petersen et al., 2015, Taveggia et al., 2005, Azzolin et al., 2014)…”

[Editors' note: further revisions were requested prior to acceptance, as described below.]

*The authors have carefully addressed all comments and questions that were raised by the referees. We acknowledge that this was a relatively long list that was contributed by several experts in the field, who now had a chance to study the revised manuscript. Before eLife can accept the paper, the authors should address a few remaining points that mainly require rewording of text.*

*1) While it has been clearly stated in the rebuttal letter that the main discrepancy between the present study and that of Poitelon et al. is due to a more thorough analysis of YAP/TAZ in adulthood, this is less evident in the manuscript itself (Results section, subsection “Yap/Taz are required for timely axon sorting”, third paragraph). Please adjust.*

We now explicitly stated in the third paragraph of the subsection “YAP/TAZ are required for timely axon sorting” that “The discrepancy … due to the fact that Poitelon et al. examined *Yap/Taz* cDKO up to P20 …whereas we analyzed them more thoroughly up to P60.”.

*2) We are not convinced that there is "better axon sorting in distal nerves of cDKO at P18" (subsection “Yap/Taz are required for timely axon sorting”, third paragraph), a conclusion based merely on low magnification semi-thin sections. Please be more cautious.*

We have removed the statement.

*3) The correlation of "weak staining in YAP/TAZ" and the failure of mutant SC to enter S phase is too speculative. Unless the expression of cyclins is provided, this conclusion should be softened (subsection “Yap/Taz are required for S-phase entry and proper proliferation of immature Schwann cells”, second paragraph).*

We have softened the statement referred by the reviewer and also amended title of the subsection (“YAP/TAZ are required for proper proliferation of immature Schwann cells”, second paragraph).

*4) There is no strong evidence of TAZ/YAP directly activating Krox20 in Schwann cells and the authors themselves stated "These results suggest that Yap/Taz directly regulate Schwann cell myelination most likely by activating transcription of Krox20" (Discussion, third paragraph). Thus, the paper's title ("Yap/Taz initiate and maintain Schwann cell myelination by upregulating Krox20") appears too strong of a statement. YAP/TAZ may regulate many other factors required for Schwann cell myelination. In the absence of evidence demonstrating the ability of TAZ/YAP to upregulate Krox20, e.g. by overexpression studies, the title should be adjusted.*

‘[…]by upregulating Krox20’ is removed in the new title.

*5) The authors added "new data" for the iDKO/PlpCreERT phenotype as a single image in Figure 6E5. However, the same image was shown in the previous manuscript (Figure 6E3 in the first submission), where it was labeled as iDKO/Sox10-CreERT. Please check and correct.*

Corrected.

*6) The authors had been asked to provide data for SC numbers, survival and proliferation in iDKO. In their response, the authors claim to have performed these experiments but do not provide data ("Lastly, as suggested by the reviewer, we have carried out a small number of additional experiments to check SC numbers and proliferation in iDKO: We have found that SCs are not reduced and that proliferation appears only marginally above background. We plan to provide these data when we report the results of our ongoing study."). For a convincing statement, the authors should at least provide a figure for the referees.*

Figure 9: The results are obtained from 3 control and 3 Plp-CreERt2-Yap^f/f^/Taz^f/f^ mice, 18 days after 1^st^ tamoxifen injection. This time point was chosen because it is just prior to the 3^rd^ week after injection, when iDKO mice begin to die. Schwann cell number is not different between WT and iDKO mice (A, see also B for SCs labeled by Sox10). We also observed neither proliferating (B; EdU staining) nor apoptotic SCs (C; TUNEL staining) in iDKO sciatic nerves.

Author response image 1.**DOI:**
http://dx.doi.org/10.7554/eLife.20982.021